# StratoMod: predicting sequencing and variant calling errors with interpretable machine learning
Nathan Dwarshuis [1]✉, Peter Tonner[1], Nathan D. Olson [1], Fritz J. Sedlazeck [2,3], Justin Wagner[1] & Justin M. Zook [1]

Despite the variety in sequencing platforms, mappers, and variant callers, no single pipeline is optimal across the entire human genome. Therefore, developers, clinicians, and researchers need to make tradeoffs when designing pipelines for their application. Currently, assessing such tradeoffs relies on intuition about how a certain pipeline will perform in a given genomic context. We present StratoMod, which addresses this problem using an interpretable machine-learning classifier to predict germline variant calling errors in a data-driven manner. We show StratoMod can precisely predict recall using Hifi or Illumina and leverage StratoMod's interpretability to measure contributions from difficult-to-map and homopolymer regions for each respective outcome. Furthermore, we use Statomod to assess the effect of mismapping on predicted recall using linear vs. graph-based references, and identify the hard-to-map regions where graph-based methods excelled and by how much. For these we utilize our draft benchmark based on the Q100 HG002 assembly, which contains previously-inaccessible difficult regions. Furthermore, StratoMod presents a new method of predicting clinically relevant variants likely to be missed, which is an improvement over current pipelines which only filter variants likely to be false. We anticipate this being useful for performing precise risk-reward analyses when designing variant calling pipelines.

The current era of sequencing offers an array of short-read and long-read technologies to identify the bases of a DNA molecule. However, different genomic contexts give rise to varying performance of technologies to measure the human genome[1]. Challenging genome contexts include repetitive adjacent bases, repeats in a local neighborhood, and large duplications. For example, homopolymers are a challenge for most sequencing technologies, and the few that do perform well in homopolymers are short-read technologies which don't have the same mapping capabilities as long-read platforms. Furthermore, the bioinformatics tools (such as read mappers and variant callers) one could employ to analyze data from these platforms also have different performance characteristics depending on the genomic context. Thus to fully take advantage of the sequencing ecosystem, it is important to understand how combinations of each platform and bioinformatics tool (ie "pipelines") perform in different genomic contexts.

Understanding how genomic context affects the accuracy of sequencing, mapping, and/or variant calling is important for a variety of applications. First, the Genome in a Bottle Consortium (GIAB) maintains a set of DNA reference materials and variant benchmarks for several well-characterized genomes[2]. These benchmarks are VCF files representing variants for each genome in comparison to GRCh37 or GRCh38, and creating these benchmarks involves leveraging the strengths of many sequencing platforms within the varying contexts throughout the human genome. Second, utilizing multiple sequencing technologies was instrumental in creating the T2T-CHM13 reference[3], the Human Pangenome Reference Consortium diploid assemblies[4], and the T2T-HG002 Q100 diploid assembly[5]. Indeed, GIAB recently leveraged the latter in creating a draft assembly-based benchmark for HG002 (which is featured later in this work). Third, many challenging medically relevant genes occur in difficult-to-sequence regions[6–8]. Therefore, designing clinical assays or designing clinical trials would benefit from understanding which technologies are most appropriate given the regions under study. In fact, the Association for Molecular Pathology recommends the use of representative variants in their bioinformatics guidelines[9], and these must satisfy FDA's requirements for regulatory submissions[10]. In each of these examples, choosing the sequencing platform and subsequent bioinformatics tooling requires making informed tradeoffs between reagent cost, time, compute power, user

[1]Material Measurement Laboratory, National Institute of Standards and Technology, Gaithersburg, MD, USA. [2]Human Genome Sequencing Center, Baylor College of Medicine, Houston, TX, USA. [3]Department of Computer Science, Rice University, Houston, TX, USA. ✉e-mail: njd2@nist.gov

expertise, and desired level of accuracy given the application at hand and the relevant regions within the human genome.

To measure performance (often either precision or recall) in these differing genomic contexts, GIAB maintains a set of bed files called "genome stratifications" which bin the human genome into different region types. As an example, the precisionFDA Truth Challenge V2 used stratifications to compare the strengths and weaknesses in different genomic contexts of sequencing technologies[1]. These stratifications encompass the following categories which are expected to have an impact on variant call accuracy: Low Complexity, Functional Technically Difficult, Genome Specific, Functional Regions, GC content, mappability[11], Other Difficult or erroneous reference regions[8], Segmental Duplications, Union of multiple categories, Ancestry of the reference[12], and sex chromosomes. By using these stratifications, we were able to show that Oxford Nanopore Technologies (ONT) reads had higher performance in segmental duplications and hard-to-map regions whereas Illumina excelled in low-complexity regions like homopolymers.

While stratifications can be useful in assessing performance, they themselves do not provide a model for where errors are likely to occur. To this end, a variety of approaches have been used to model sequencing errors, mostly as part of variant calling to filter variants from the callset which do not exist in the genome being measured. For example, GATK Variant Quality Score Recalibration uses Gaussian Mixture Models to identify abnormal read characteristics[13]. More recently several deep learning models have been developed to minimize the need for expert-curated features by taking in sequences from the reference and characteristics of aligned reads in a small region around each candidate variant[14–17]. Additional methods have been designed by clinical laboratories to predict which variants in a callset are likely to be real and do not need to be orthogonally confirmed by another method like Sanger sequencing[18,19]. All of these methods have been very useful, particularly for increasing variant calling precision, but they have important limitations. For example, deep learning and many machine learning methods lack interpretability, and all the above methods focus mostly on sequencing read characteristics at the expense of genome context, and they do not predict true variants that are likely to be missed.

In this work, our goal was to develop an interpretable model to predict the precision and recall for calling a variant using a specified method given its genome context. Interpretability was desired to allow end users of our model to understand how each feature (which corresponds to an aspect of genome context) contributes to a prediction[20]. To this end, we chose Explainable Boosting Machines (EBMs)[21], which are a specific implementation of generalized additive models (GAMs) where model predictions are derived from additive effects of univariate and pairwise functions of dataset features (see "Methods" section for equation form). Each of these functions can be plotted individually to assess its impact on the response. EBMs have previously been shown to identify patterns in data that were obscured by other models, including confounding effects that can only be explained by domain experts[22]. In our use case, this aspect is especially important for clinicians who generally are required to justify their decisions to patients or other stakeholders (beyond simply saying "the model told me").

This modeling approach using EBMs with genomic context features, which we call StratoMod, offers several advantages over the current strategy for assessing performance based on GIAB stratifications. First, StratoMod is much more precise. In the case of homopolymers, the genome stratification approach would have required a decision to threshold discrete bins of homopolymer lengths such as 4–6 bp, 7–10 bp, >10 bp, and >20 bp; in this case, errors can only be reported in terms of these discrete bins. In contrast, the EBM model reports errors in terms of a continuous scale (log-odds), in which users can more precisely identify the impact of homopolymer length on the likelihood of an error which in turn can highlight biases or strength of different sequencing technologies. Second, this model approach allows multiple genome contexts to be assessed simultaneously. Since EBMs can also include bivariate terms, we can inspect the interaction between homopolymer and INDEL length for instance. Since INDELs themselves

can have varying difficulty depending on their length, sign (e.g., insertion or deletion), and method by which they are measured, it would be useful to understand how homopolymers (or other genome context) modulate this difficulty. This would also be important for assessing structural variants (INDELs >50 bp), which we did not address in this work but are nonetheless of interest to the field. Third, StratoMod can predict both precision and recall for a given method. This is a step forward for the field because many variant calling pipelines will filter candidate sites with poor support whose inclusion would likely hurt precision. However, the inverse is not true; current variant calling pipelines have no way of adding missing variants that are true, which would improve recall. While GIAB has previously created stratification bed files which include hard-to-map and segmental duplication regions which in theory are enriched for variants likely to be missed, these bed files provide no predictive quantification on their own. StratoMod solves this limitation by using well-characterized benchmarks as a source of truth, and inferring where a variant will likely be missed based on the context of each variant.

## Results

Figure 1 shows the overall approach of StratoMod. For this study, we trained multiple iterations of StratoMod to showcase its ability to predict recall, precision, or Jaccard index (Fig. 1c). In the sections "Use case 1", "Use case 2" and "Other use cases" below, we use StratoMod to predict recall of several pipeline configurations. We focused on recall in this study since this is currently difficult to predict for pipelines, and thus showcases StratoMod's value. We also trained StratoMod to predict precision in Supplementary Note 2. In "Model Validation" below we use StratoMod to predict the Jaccard index ($TP/(FP + FN + TP)$) for variants in ClinVar, which we compare to the likelihood of matching variants in gnomAD being filtered. Here we predicted the Jaccard index instead of precision or recall since a filtered variant in gnomAD may correspond to either a false positive or false negative.

In each case, we trained two individual models for INDELs and SNVs separately. Furthermore, all false positive (FP) and false negative (FN) errors were determined with respect to either the GIAB v4.2.1 benchmark VCFs or our new T2T-HG002-Q100 assembly-based benchmark (see "Methods" section) using GRCh38 as the reference. Lastly, we simplified the analysis by splitting multi-nucleotide variants, removing SVs (variants larger than 50 bp), genotype errors, and any variants that appeared in the MHC regions from both the benchmark and query VCF input files.

### Use case 1: predicting recall in Illumina PCR-free and PacBio HiFi variant calling pipelines

We first asked if StratoMod could be used to assess the likelihood of missing a variant given its genomic context and the read length used to assess the variant. To this end, we trained StratoMod using our new draft assembly-based benchmark for HG002, which is based on the near-perfectly complete Q100 HG002 assembly[5] and thus allows us to assess variation in some of the most difficult-to-analyze regions of the genome (Fig. 2a). Furthermore, we used the trained models to predict recall for pathogenic/likely pathogenic ClinVar variants, which should provide useful insight to diagnostic test developers and clinicians who may be concerned about the likelihood of missing a potentially lethal variant in a patient.

**Identifying driving features that influence recall.** We trained separate models for SNVs and INDELs using VCFs generated using DeepVariant with GRCh38 as the reference and reads generated from HG002 using either Illumina PCR-free or Pacbio Hifi. Each model was trained on false negatives (FN, defined as variants in the benchmark that are missing or have an incorrect allele or genotype in the query)[23] and true positives (TP, defined as variants matching a benchmark variant and genotype) variant call classifications, as reported by vcfeval using our Q100 HG002 assembly-based benchmark as the truth set. The model utilized 22 main effect (univariate) features, including 1 categorical feature denoting the sequencing platform (Hifi or Illumina). These main effect features

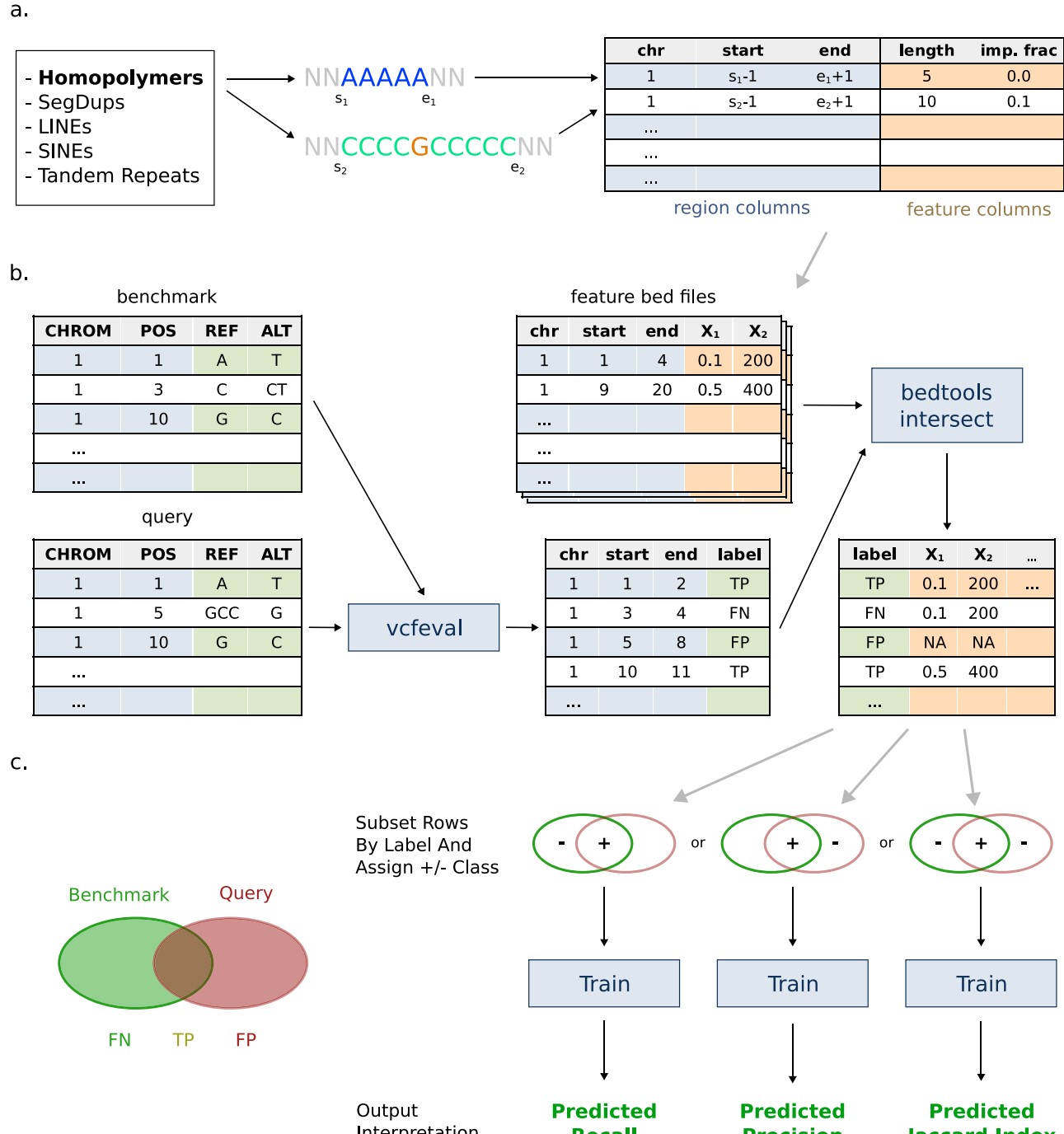

**Fig. 1 | Graphical overview of StratoMod. a** Conceptual framework for mapping "genomic context" to a machine-understandable value (with homopolymer length as an example). **b** Flow chart for the analysis pipeline, where a query callset is compared to the benchmark to identify TPs, FPs, and FNs. These are then intersected with regions and metadata describing different genomic contexts, producing a mapping between variant calling results (TP, FP, FN) vs genomic context which are used as labels and features in the model respectively. **c** This data frame is then filtered for different labels and is assigned positive ("+" for TP) or negative ("−" for FN and/or FP) class depending on the desired prediction. Subsetting to the benchmark variants (FN + TP), query variants (FP + TP), or both (FN + FP + TP) will predict recall, precision, or Jaccard index respectively. SegDups Segmental Duplications, LINE long interspersed nuclear element, SINE short interspersed nuclear element.

included quantified characteristics of homopolymers, tandem repeats, segmental duplications, and other repetitive elements, many of which we have seen cause mapping errors and/or sequencing errors which will lead to missed variants. Each model also included interaction terms between the 22 main effects and the Hifi/Illumina categorical feature, allowing the model to show the behavior of each main effect conditional on technology (see "Methods" section and Supplementary Table 1 for a complete list of features).

We show two examples of such variants that might be predicted using StratoMod in Fig. 2b. The left panel depicts a variant in a LINE (long interspersed nuclear element) that Illumina missed but Hifi called correctly, presumably because long reads were able to map correctly to the LINE. The right panel depicts a homopolymer where the variant was called correctly from Illumina data, but DeepVariant incorrectly classified the candidate variant as a homozygous reference from HiFi data, presumably because HiFi data is noisier due to the homopolymer.

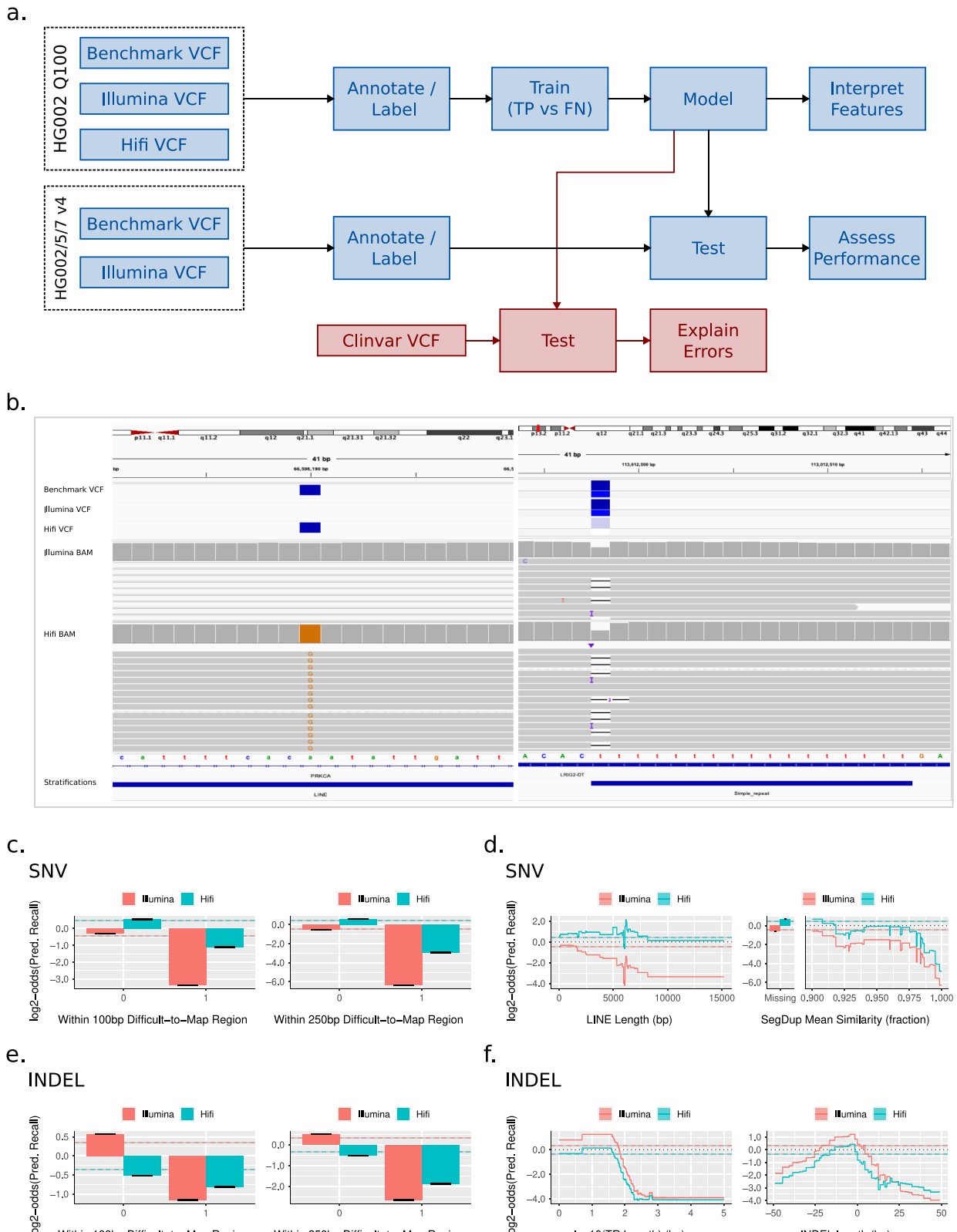

**Fig. 2 | StratoMod found context-specific regions where variants are likely to be missed in either HiFi or Illumina analysis pipelines. a** Experimental overview **b** IGV images depicting false negative calls identified by this model. **c–f** EBM interaction feature plots for SNV hard-to-map regions (**c**), SNV LINEs and segmental duplications (**d**), INDEL hard-to-map regions (**e**), and INDEL tandem repeats/indel length (**f**). LINE long interspersed nuclear element. Error bars and ribbons around step plots are model errors.

StratoMod enables us to systematically quantify error likelihood for variants such as these using feature profiles (Fig. 2c–f). Each profile is actually the addition of three features output by the model: the main effect of the sequencing platform (color), a region type such as tandem repeat length (x-axis), and the interaction of the two (divergence of the two colors in the profile). The y-axis is the log2 odds ratio of the predicted recall (i.e. an increase in 1 corresponds to twice as likely to find a true variant). For SNV and INDELs, we found that hard-to-map regions were unsurprisingly predictive of higher error rates (Fig. 2c, e). Furthermore, the increase in error rate was less for Hifi than Illumina, likely due to longer read lengths provided by Hifi assisting in mapping. For SNVs, we also observed that variants in a LINE were more likely to be missed than those not in LINEs, and the difference in predicted score between Illumina and HiFi became larger as length increased (Fig. 2c). There was a spike in the FN rate around 6000 bp long (corresponding to full-length L1 LINEs), possibly because full-length LINEs are more recent and similar to each other. Interestingly, segmental duplications for SNVs did not show a wide divergence between Illumina and Hifi, although the error rate increased with increasing similarity. This could be partly explained by the difference in the difficult-to-map effect in Fig. 2c between Hifi and Illumina, which overlaps with the segmental duplications effect in Fig 2d (as segmental duplications are also hard-to-map generally). For INDELs we observed that increasing tandem repeat length generally increased error rate, and Hifi's error rate decreased slower, in particular as repeat length exceeded the length of a short-read (100 s) (Fig. 2f). Furthermore, long deletions and insertions generally had higher error rates, and Hifi had a lower error rate relative to Illumina for long insertions, likely due to the fact that mapping becomes harder as insertions become longer. While these only represented a subset of all features tested, they represent many of the high-leverage features that contribute to the resulting prediction (Supplementary Fig. 1a). In total, these profiles support what we already know about short vs long reads given our intuition about mappability and long repetitive regions; however, they further enable better precision for understanding both the error rate and the types of regions that lead to a given error rate. An important caveat is that these profiles can change substantially depending on the mapping and variant calling methodology, as we show below for linear vs. graph-based references.

**Performance assessment and prediction of missing clinically relevant variants.** Each model was trained using an 80/20 test split on HG002 (Ashkenazi Jewish ancestry) (Fig. 2a); we additionally tested the model on HG005 and HG007 (Han Chinese ancestry) to assess its generalizability to other genomes. Since we did not have an assembly-based benchmark for these, we used our v4.2.1 benchmarks instead and also tested the v4.2.1 benchmark for HG002 which we expected would give us similar results to the assembly-based HG002 benchmark (which is mostly a superset of the v4.2.1 benchmark). We observed that both precision and recall (note these two metrics refer to the model's classification performance and are distinct from the predicted precision and recall which describe StratoMod's output) were generally similar and high between all models and genomes (Supplementary Fig. 1b). SNV performance for HG002 Q100 was near 1.0 for the area under the precision-recall (PR) curve and about 0.96 for the area under the receiver-operator (ROC) curve for both Illumina and Hifi (Supplementary Fig. 2d). INDEL performance was lower (PR = 0.84–0.86 and ROC = ~0.985). These metrics were similar for HG002 between the two different benchmarks, with Illumina INDELs being slightly higher for v4.2.1. The other genomes were also similar, again with the exception of Illumina INDELs were slightly more performant. We also noted that when stratifying the ROC and PR curves by platform and variant type, the curves overlapped almost perfectly, indicating the performance across the two platforms is more-or-less equal (Supplementary Fig. 2e). Together these indicate that the models are well-trained and that feature profiles are describing trends in the data reasonably well.

**Assessment of pathogenic ClinVar variants.** We next assessed the ability of our model to predict the likelihood of missing clinically relevant variants with either Illumina or Hifi reads. We fed a ClinVar VCF through our model and extracted the probabilities of missing each variant. We also examined the contribution of each feature to those probabilities to "explain" why some variants were likely to be missed. To focus on variants of more clinical interest, we only considered variants marked as "pathogenic" or "likely pathogenic" in the ClinVar VCF. We then bisected variant predictions to those greater or less than 90%, with the former being deemed "detected" and the latter "missed". The models were well-calibrated around this 90% cutoff, indicating that our cutoff should roughly correspond to the real-world likelihood of missing a variant (Supplementary Fig. 1a). Note that the vast majority of variants were above this 90% threshold (Supplementary Fig. 2b, c) thus the variants below this cutoff represented the lower tail of the hardest variants to call.

We then plotted missed variants where Illumina and Hifi predictions disagreed (Fig. 3a) and agreed (Fig. 3b) with each other along with the relative contributions from each feature. We found that Hifi had many fewer variants lower than our 90% cutoff for both INDELs and SNVs (Fig. 3a). Furthermore, variants for which Illumina and Hifi disagreed had different features driving their predictions. Illumina's predictions were largely driven by segmental duplications, mappability features, tandem repeats, and indel length (in the case of indels). The few errors in Hifi by contrast were mostly in homopolymer regions. For cases where Illumina and Hifi both missed variants, the feature contributions were largely similar (Fig. 3b). We also noted that Hifi tended to assign higher probabilities overall, particularly in the case of SNVs where Hifi was as confident as Illumina or more but the reverse was not true (Fig. 3c). For INDELs this was also largely true except for SNVs >90% where Illumina was sometimes more confident than Hifi. Together these indicate that while Hifi is overall much less likely to miss variants (particularly in hard-to-map regions) when using DeepVariant with a standard linear reference, it still might miss errors in homopolymers that would otherwise not be missed with Illumina.

Most predicted missed pathogenic and likely pathogenic variants were explained both by low mappability and being in a highly similar/duplicated segmental duplication, with more than 20 FN SNVs and INDELs predicted for Illumina-DeepVariant and/or HiFi-DeepVariant in genes PKD1, *PMS2, NF1, CYP21A2, CHEK2, NEB, FLG, SDHA, ABCC6, SMN1, STRC*, KMT2D, CYP11B1, CDKN1C, *PIK3CA*, PROSS1, HYDIN, and TUBB2B, as well as genes that are falsely duplicated in GRCh38 (CBS, with fewer FNs in *KCNE1, U2AF1*, and *CRYAA*), with full list of genes and counts of predicted FNs in Supplementary Data 1 and Supplementary Note 1. This is plausible given that these two region types should overlap significantly and also should lead to mismapped reads and hence missed variant calls. Interestingly, HiFi-DeepVariant had at least 5-fold fewer predicted missed variants in all of these genes except CBS, *CYP21A2, NEB, ABCC6, and HYDIN*, which were predicted to be challenging for both technologies. However, this finding may not apply to all methods, such as methods that mask the falsely duplicated regions in GRCh38 that cause mapping challenges in CBS and other genes.

In addition to segmental duplications, some variants predicted to be missed were related to INDEL length, tandem repeats, and other difficult-to-map regions, indicating that multiple error mechanisms may be at play. For example, potentially missed variants were predicted for INDELs in VNTRs in *ACAN, COL6A1, F7, MYO7A, AGRN*, and *CDKN1C* (some of which are inside an exon and some partly exonic but mostly intronic), as well as large insertions in trinucleotide tandem repeats like *DMPK, PHOX2B*, and *RUNX2*, where expansions are associated with disorders. The remaining predicted missed variants were large INDELs in non-repetitive regions (a few incidentally in homopolymers), except for a cluster in a few hundred bp regions of *ANKRD11* that is identical to a region on chrX but can be mapped accurately with most paired-end reads. The full list of genes and counts of predicted missed variants is in Supplementary Data 1.

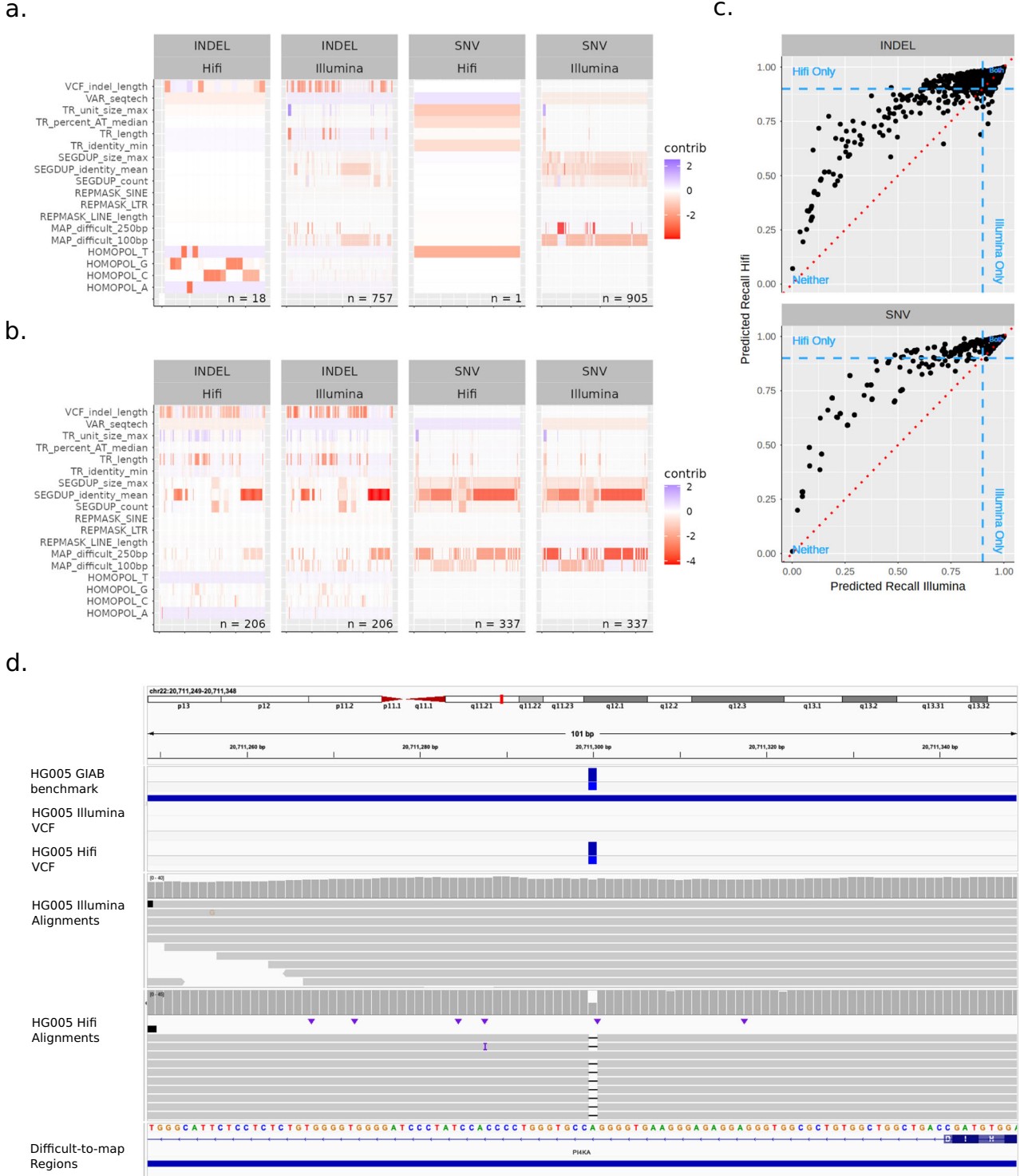

**Fig. 3 | Assessment of ClinVar variants predicted to be missed using Illumina and HiFi calls from DeepVariant. a, b** Heatmap of variants with a 10% or higher likelihood of being predicted as missed unshared between Illumina/Hifi (**a**) or shared (**b**) showing contributions of each feature within the model. Note that all interaction terms were added to their univariate main effect features to simplify visualization, and also note that negative scores contributed positively to a missed variant. **c** Predicted probabilities between Illumina and Hifi models. Red dotted red line is equal probability for Illumina and Hifi. Blue dotted lines are 90% probability cutoffs for either platform **d** ClinVar variant that is correctly predicted to be missed in Illumina-DeepVariant and as a TP in HiFi-DeepVariant due to being in a difficult-to-map region in a segmental duplication in *PI4KA*.

Many of the above-identified genes, tandem repeat regions, and variant types have previously been known to be challenging. In addition to highlighting general genes and characteristics of variants, this model predicted particular pathogenic variants that may be missed by a particular sequencing and bioinformatics method (in this case, 35× Illumina-DeepVariant PCR-free WGS), which may not be representative of all Illumina-based methods. This model could be used similarly to predict variants that might be missed by any

method that has been used to call variants from benchmark samples like those available from GIAB.

## Use case 2: comparing linear vs graph-based mappers

We next hypothesized that StratoMod's feature set would enable it to differentiate the performance of different mappers, particularly linear and graph-based mappers because graph-based mappers have been developed to improve results in difficult-to-map regions and variants[24,25].

To test this, we trained StratoMod using two Element DeepVariant callsets which used either BWA mem[26] (linear) or VG[27] (graphical) mappers prior to running DeepVariant. The model was trained using TP as the positive class and FN as the negative class and thus learned to predict where variants were most likely missed. Between VG and BWA, the models were generally similar except for a few key features that corresponded to hard-to-map regions, and the performance/calibration of each model was reasonable (Fig. 4a, b, Supplementary Fig. 3a). As expected, VG's error rate increased less than BWA's when moving from non-hard-to-map vs hard-to-map

(either 100 or 250 bp long) for both SNVs and INDELs. Furthermore, VG's error rate increased less with increasing LINE length for both SNVs and INDELs, corresponding to the intuition that longer LINEs should be harder to map. Finally, in the case of INDELs BWA's error rate dropped when similarity increased above 92% while VG dropped less at the same point, again corresponding to the intuition that more similar segmental duplications are harder to map. Each of these depicted features was representative of a key subset of features whose main effect and interaction with the two mappers contributed to a large component of the model's variability (Supplementary Fig. 3b).

We next asked what distinguished the variants for which the two mappers disagreed. We plotted StratoMod's predicted recall for cases where BWA was correct and VG was incorrect and vice versa (Fig. 4c, Supplementary Data 3, 4). These probabilities were reasonably calibrated for both SNVs and INDELs (Supplementary Fig. 3c). Interestingly, in either case using VG often led to higher predicted recall than BWA (by as much as 50%), even when it was incorrect according to our benchmark. The reverse

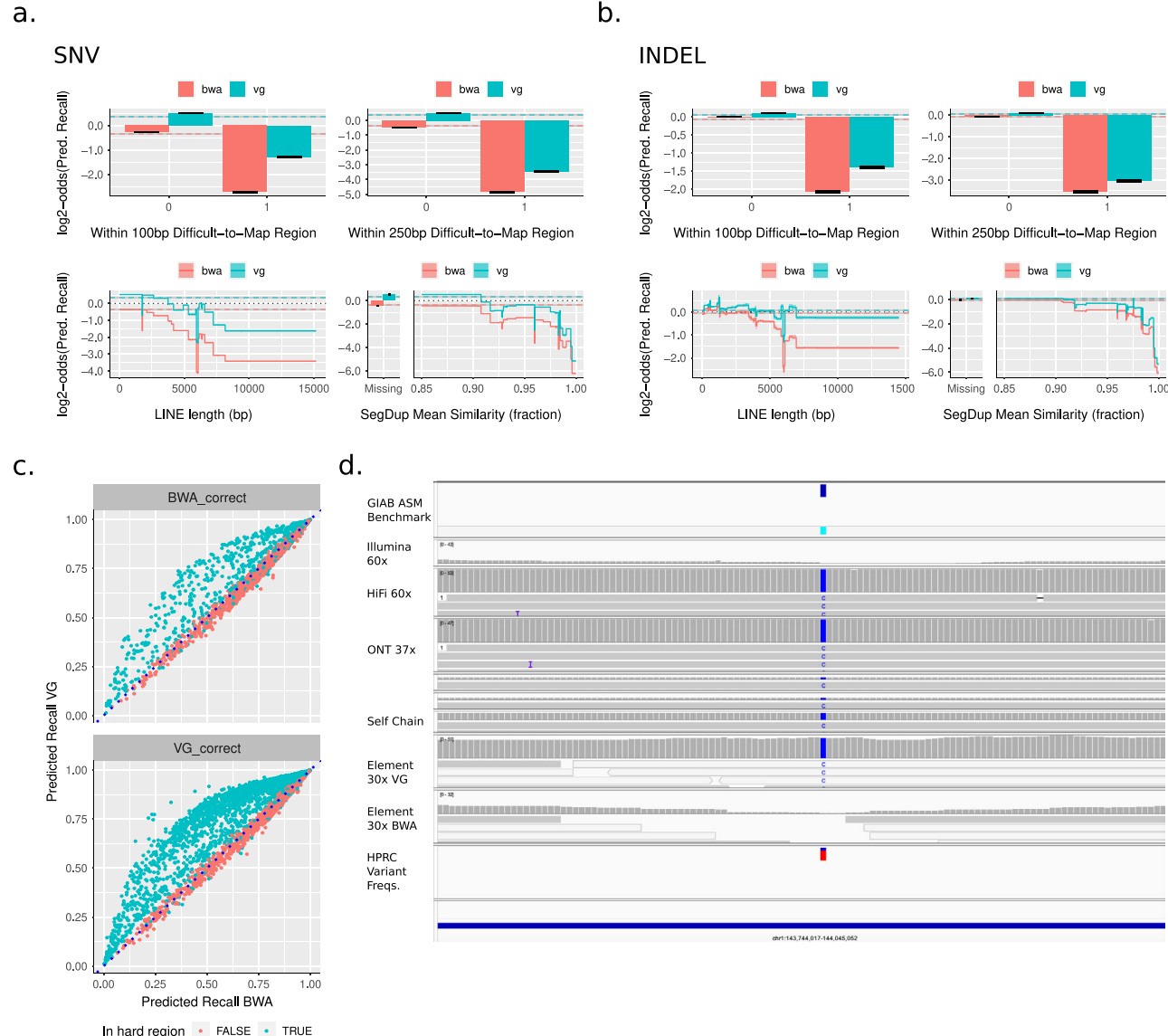

**Fig. 4 | StratoMod correctly predicts the advantage for VG in hard-to-map regions. a**, **b** Prominent hard-to-map feature profiles. Error bars and ribbons around step plots are model errors. **c** Prediction correlation plot between VG and BWA where either BWA or correct and VG is incorrect (top) or the reverse (bottom). Dotted line y = x (perfect correlation). Note these are only for SNVs but the INDEL plots were almost identical. "Hard regions" included LINEs, SINEs, LTRs, segdups, and hard-to-map regions 100 and 250 bp long **d** IGV screenshot depicting an instance where VG is correct and BWA is incorrect. In this case, the variant appears in this copy of the segdup but not others for a large fraction of haplotypes in the graph (bottom track).

(using BWA lead to higher predicted recall than VG) appeared to never be true. When stratifying these plots by those in "hard" regions (LINES/SINES (short interspersed nuclear element)/LTRs (long terminal repeats)/Segdups/Hard-to-map 100 and 250 bp long), we saw that this increased predicted recall was due to VG performing better in these particular region types.

We next manually investigated cases where the two mappers disagreed, and highlighted a representative case where VG was correct and BWA was incorrect due to a segmental duplication (Fig. 4d). This variant was present in HG002's genome (top benchmark track) but the self-chain alignment of the other copy of this segmental duplication to this region shows that it matches this variant, causing Element reads with the variant to map to the wrong copy using a linear reference. However, because the variant is also present in most of the HPRC assemblies in the graph (bottom track), VG was able to leverage this graph to accurately map reads to this region rather than the other copy of the segmental duplication.

Taken together, these demonstrate how StratoMod can be used to compare and assess the performance of different mappers given genomic regions with different mapping characteristics.

## Other use cases

**Assessment of new sequencing technologies.** New sequencing technologies are under active development, and methods to understand the strengths and weaknesses of these technologies are important, so we next tested if StratoMod could be used to measure the progression of Ultima's new sequencing platform that promises to be less expensive and has been steadily improving up to their first product launch in 2024[28]. We compared their latest callsets ("R2024") from the UG100 to those from 2022 ("R2022"), both called by DeepVariant. These models were trained analogously to those in use case 1 (same feature set, benchmark, and genome) and were trained to predict recall.

We particularly investigated homopolymer errors since these features were dominant in the model's predictions (Supplementary Fig. 4a, b). For AT homopolymers, the error rate climbed precipitously after 8 and 10 bp for SNV and INDEL respectively. For GC homopolymers the error rate rose almost immediately after 4 bp (which was our minimum for defining a homopolymer). Furthermore, Ultima's latest iteration had notably higher overall accuracy compared to their previous iteration, as noted by the translational increase in the R2024 curve relative to the R2022 curve in each of the plots and the "VAR_seqtech" variable (which is the categorical variable distinguishing the two releases) being near the top in the feature importance plots (Supplementary Fig. 4b). Subtly, their newer iteration showed a slight advantage for longer homopolymers (particularly INDELs) where the distance between the curves grew with increasing homopolymer length.

In total, these data demonstrate how StratoMod can be used to assess the progress of emerging technologies; in this case, the overall error rate dropped considerably for the latest Ultima iteration, with subtle performance differences conditional on region type (such as long homopolymers).

**Assessment of variant calling pipeline improvements.** In addition to the platforms themselves, the software used to call variants from data created on these platforms is constantly improving. We used StratoMod to assess improvements in predicted recall for different versions of guppy and clair, the base and variant caller respectively for the ONT sequencing pipeline. Specifically, we assessed guppy4+clair1 and guppy5+clair3 in combination. These were trained analogously to use case-1 except that we used HG003 as the benchmark.

We again investigated homopolymer errors since this is a well-known error modality for ONT callsets. Overall error rates were much lower for the newer software as expected (Supplementary Fig. 5, Supplementary Note 3) with the improvement being much more pronounced for INDELs. In the case of SNVs, much of the performance improvements were independent of homopolymer length and imperfect fraction. However, the INDELs, the predicted recall for the older callers degraded much more rapidly for

homopolymers longer than 8 bp and 6 bp for AT and GC respectively (Supplementary Fig. 5a). The gap between the older callers and the newer callers decreased with increasing imperfect fraction, especially for AT homopolymers (Supplementary Fig. 5b). Despite this, the older callers were still inferior to the newer callers by at least a 5x margin for each imperfect fraction. These suggest that the homopolymer-specific error mechanism in the older ONT callers (which the newer callers improved) was due to miscounting longer homopolymers. This would affect INDELs more than SNVs since vcfeval will only count a variant if it matches completely. This also would be somewhat negated in homopolymers with higher imperfect fractions since imperfect bases "interrupt" a stretch of otherwise similar bases as they are going through the nanopore, which may make counting the number of bases easier. In fact, one approach ONT has announced to improve accuracy in homopolymers (the 6b4 chemistry) is to modify bases inside the homopolymer.

Together, these indicate how StratoMod is able to assess improvements in variant/base callers and provide insight into the mechanism for these improvements.

## Model validation: comparison between ClinVar predictions and gnomAD

We finally sought to validate the use of our model's predictions of ClinVar variants that may be missed or erroneously called by short reads. Given that no benchmark exists for most ClinVar variants from which we can derive "labels" (TP, FP, etc), we hypothesized that the probability of our model should be correlated with the likelihood of a variant in gnomAD v4.0 being filtered. gnomAD may filter a variant for similar reasons that StratoMod may assign a low probability, albeit with a different model and different input data. We trained similar models to those used in "use case 1" except we also included FP (along with FN) as the error label since errors in gnomAD could correspond to either an FP or an FN error in StratoMod (note that this is the Jaccard index, or TP/(TP + FN + FP) (Fig. 1c). Previously we limited ourselves to FN to demonstrate StratoMod's ability to predict recall. We also trained StratoMod using an Illumina callset to be comparable to gnomADs underlying sequencing data, albeit with different variant calling methodologies (both used BWA mem, and the training set used DeepVariant). Thus the interpretation of StratoMod's output is "the probability that ClinVar variants of interest will be correctly called using Illumina-BWAMEM-DeepVariant." Many ClinVar variants did not intersect with any gnomAD variants (Supplementary Fig. 13a); since many missing variants are likely very rare and not in any gnomAD samples. While some of the missing variants are in challenging regions, many are not, so the rest of the analysis only concerns the fraction of ClinVar variants that intersected with either a filtered or passing gnomAD variant.

We first created "agreement plots" (similar to calibration plots in machine learning) which depict the binned fraction of gnomAD non-PASS variants vs the mean predicted error rate (which is 1 - StratoMod's output) for that bin (Fig. 5a). In general, StratoMod and gnomAD corresponded reasonably well, with StratoMod predicting slightly fewer errors than gnomAD's filtering (i.e. assigned a lower probability than the fraction of non-PASS variants) for the majority of variants for both SNVs and INDELs. Note that for both, the curves were noisier moving toward the lower-left corner since the number of variants in each bin decreased with decreasing probability. To explain why Stratomod predicted fewer errors, we further hypothesized that gnomAD is filtering true variants with low allele count (AC) in a region type that StratoMod predicted would be easier than the number of filtered variants would suggest. We tested this by stratifying the agreement plots by allele counts greater than/less than 10 (red vs blue lines) and indeed noted that for the majority of variants, StratoMod predicts fewer errors for low allele counts.

We further tested this hypothesis by curating some of the low allele count variants in the gnomAD browser. We found that these variants mostly fell in two categories: (1) variants in only one or two whole genome samples that did not have evidence of mapping or sequencing errors so appeared likely to be true, and (2) variants in homopolymers that were likely

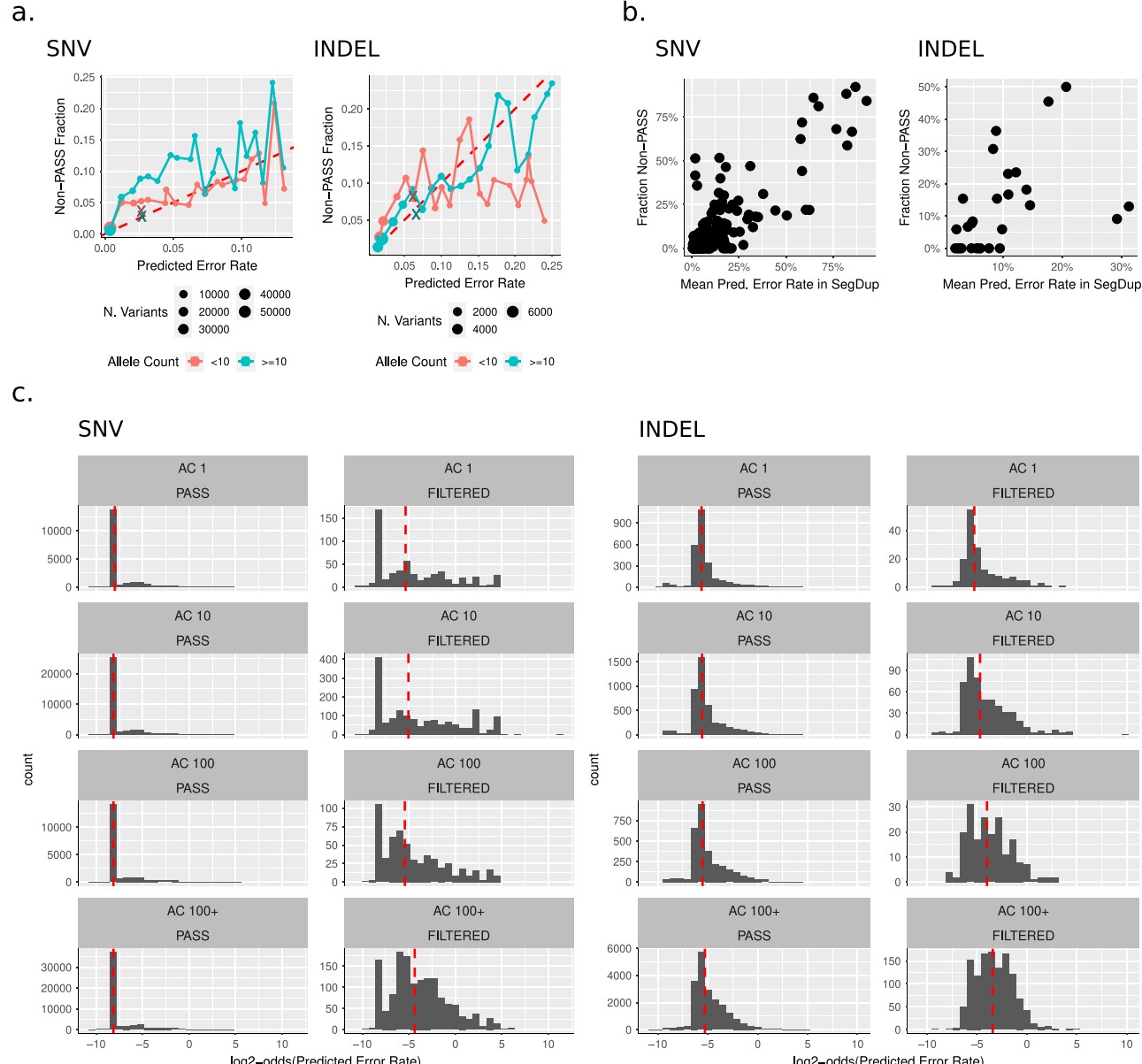

**Fig. 5 | Validation of ClinVar model predictions using gnomAD. a** Agreement plots between model predicted error rate and fraction of gnomAD non-PASS variants using 20 bins and stratified by gnomAD allele count. Each dot is a binned model probability, where the x position is the mean probability and the y position is the fraction of non-PASS in that bin. The "x" points are all variants that had no features in the model (and thus their probabilities are nearly identical) and the remaining points are variants with at least one feature. Red dotted line is y = x (perfect agreement). Data shown does not include the bottom 5% of probabilities, as these were sparse enough to appear as noise. **b** Correlations plots between mean model probability and a fraction of non-PASS variants in segmental duplications (SegDups). Each point represents one SegDup region and only regions with >10 variants are included. **c** Probability histograms for gnomAD PASS and non-PASS variants stratified by allele count (shown as an order of magnitude, so "AC 10" means "allele counts between 10 and 99"). Dotted red line is the median model probability for each histogram.

sequencing errors because they only occurred in one direction (Supplementary Fig. 14). The latter variants were likely predicted to have a low error rate because StratoMod features do not include characteristics of the reads such as strand bias. Furthermore, our model is designed to distinguish between true variants and errors in Illumina-DeepVariant calls, whereas gnomAD's GATK VQSR-based filtering is designed to remove likely false variants in population callsets from 100,000's of individuals. In summary, these curated examples explain nuanced differences due to rare variants and feature sets used, but the model is generally consistent with gnomAD filtering despite these differences, as shown in Fig. 5a.

Furthermore, the "x" points in the plots denote variants that intersect with no features corresponding to challenging regions in StratoMod (or only

INDEL length in the case of INDELs), and thus have nearly the same probability. Surprisingly a large number of variants with features had higher predictions than those without, which was likely due to our feature set not including all possible features that could be used to measure "difficulty", and the inclusion of some features correlated with higher rates of true variants like short homopolymers (Supplementary Note 2 and Supplementary Fig. 15). These also followed the same pattern, with lower allele counts corresponding to StratoMod being more overconfident.

Note that in the case of SNVs, the majority of variants were in the lower-left corner (note the large dot, denoting most of the variants are in this range), which shows gnomAD had more filtered variants than predicted by StratoMod for AC < 10 vs ≥10, in line with that for INDELs (albeit with less

magnitude). However, for SNVs with probabilities between 0.025 and 0.075, gnomAD had fewer filtered variants than predicted. We hypothesized that this shift reflected a confound in the type of region represented between these different allele count stratifications and within these probabilities. To test this, we assessed the fraction of each region type (tandem repeat, segmental duplication, etc) in each bin and noted that for SNVs, segmental duplications are over-represented for variants with AC < 10 vs ≥10 for the range of probabilities between 0.025 and 0.075 (Supplementary Fig. 13b). Because StratoMod assigns a lower probability to variants in segmental duplications vs those not, this explains why gnomAD has more filtering than predicted for the AC > = 10 curve (i.e. StratoMod thinks these variants are "easier" as they are not in segmental duplications) vs the AC < 10 curve despite gnomAD more aggressively filtering variants with low allele count.

We also assessed the fraction of gnomAD non-PASS variants in segmental duplications vs StratoMod's predicted error rate (Fig. 5b). We observed a good correlation between gnomAD and StratoMod for both INDELs and SNVs. Note that in these plots we are only showing segmental duplications with 10 variants or more (regions with few variants tended to be noisier since they have fewer variants from which to estimate the x and y mean). In agreement with the segmental duplications discussed above in Fig. 5a, in the case of SNVs we observed several segmental duplications for which StratoMod predicted fewer errors compared to gnomAD in the 90–100% range. Furthermore, unlike Fig. 5a which only depicts the top 95% of variants (otherwise bins in the lower probability range would be too noisy given how little data they would contain), Fig. 5b shows variants in segmental duplications for which the StratoMod probability was less than 80%, and indeed shows that even for these lower ranges, there was high agreement between StratoMod and gnomAD.

Finally, we further assessed the relationship between allele count and StratoMod's probability by plotting histograms stratified by PASS/non-PASS and increasing allele count (Fig. 5c). We observed that probability is near 1 for the vast majority of variants in the PASS case and not so for the non-PASS case, in agreement with Fig. 5a, b. Furthermore, the median probability (dotted red line) appears to decrease with increasing allele count. This effect is very subtle in the case of SNVs (dropping by ~2x log-odds from the lowest to highest) and much higher in the case of INDELs (~8x drop from lowest to highest), also in agreement with the apparent magnitude between low and high allele counts between SNVs and INDELs in Fig. 5a.

## Discussion

In this work, we demonstrated that an inherently interpretable model with expert-designed features for genome context can be used to gain a deeper understanding of sequencing and variant calling errors.

First we demonstrated StratoMod's ability to not only predict recall for clinically relevant variants but also provide insight into what types of regions and platforms contribute to this prediction. Particularly, we showed that Hifi missed fewer variants in hard-to-map regions compared to Illumina, but the variants that Hifi missed which Illumina did not miss were in longer homopolymers. The intuition that long reads generally provide superior mapping ability over short reads is not surprising or new, but StratoMod quantifies this intuition in a way that has not been done previously and provides additional nuance depending on region type. Furthermore, the fact that these models were well-calibrated shows that they can inform key decisions and risk analysis in the context of designing lab tests or clinical trials. Analogous to the example we demonstrated, StratoMod can predict the recall of difficult variants given a platform, region type, and threshold. As new technologies such as long reads enable increasingly challenging pathogenic variants to be added to ClinVar, StratoMod could become increasingly useful for understanding the strengths and weaknesses of different pipelines for detecting these variants.

StratoMod's ability to predict recall also allows one to perform risk analyses. We demonstrated this by showing where variants may be missed given a 90% cutoff in StratoMod's predicted recall probability. Note that this cutoff was somewhat arbitrary, and its precise value must be determined by the end user after assessing the costs and benefits of each decision in their

application. Increasing the cutoff will increase the minimum confidence that a true variant would be detected, which will decrease the risk of missing a variant but also increase the number of potential FN variants. Also, note that StratoMod's "predicted recall" is related but not identical to the "probability of a false negative in any genome" which needs to consider the likelihood of the true variant existing in the genome under study (see Supplementary Note 4 for additional details).

We validated the results of this analysis by comparing the output of StratoMod's assigned probabilities for ClinVar variants in gnomAD, particularly the likelihood of gnomAD to be passing or filtered. Given the size of gnomAD, the similarity of the input data (Illumina short reads), the fact that many ClinVar variants are also in gnomAD, and the differences in the underlying models (StratoMod vs BWA/gatk), we reasoned this would provide a natural orthogonal comparison to justify StratoMod's output for ClinVar variants because GIAB benchmarks contain insufficient clinically relevant variants to test for ground truth. In general, gnomAD and StratoMod seemed to correspond quite well, at least for the majority of variants where the assessment we used is most robust. StratoMod was generally more confident (ie higher StratoMod probability vs likelihood of gnomAD passing) which seemed to be more pronounced in variants with low allele counts (less than 10) and overall was probably due to the more conservative model gnomAD entails when variants occur in few samples.

Similarly, we provided an analogous example using different types of mappers (linear and graph-based, represented by BWA and VG respectively). While the common intuition says that graph-based mappers should provide an advantage in hard-to-map regions and larger INDELs, StratoMod again provides additional insight by quantifying how much and where this intuition applies (for example, VG is ~2x less likely to miss INDELs in segmental duplications with >92% similarity). For tool developers, StratoMod provides a context-dependent lens through which their tools may be evaluated, and it also provides a means to assess the current landscape for opportunities in developing new tools to tackle difficult regions for which no good tool exists. For end users, this provides a method by which the risk-reward tradeoff of using a particular tool for a given task may be evaluated; in this example, giraffe-VG generally has a higher memory cost than BWA[24], and this may or may not be important depending on the goal.

We selected a relatively small set of features to maintain interpretability and reduce correlations between related features as this reduces interpretability. We also evaluated a limited number of interactions between features, because large numbers of feature interactions are challenging to interpret. We expect adding and fine-tuning features and adding more interactions could improve model performance, but would also create more challenges in interpretation and visualization of results. The tree-based modeling approach enabled us to identify some discontinuities in scores that point to possible improvements in feature design, such as identifying peaks in segmental duplication length related to particular segmental duplications with errors in GRCh38, and a peak in LINE length related to full-length LINEs. However, the tree-based nature of the model also makes robust uncertainty estimates challenging relative to other GAM-based approaches. These results also provide suggestions for features that could be added to other models like those used by variant callers to remove false variant candidates.

While we have demonstrated that StratoMod can be useful for diagnosing and understanding variant calling errors, it was not intended to be a replacement for any existing variant caller. We trained StratoMod on FP errors for PCR-free/plus Illumina callsets, and unlike the FN models, included all candidate variant sites to test the degree to which either library prep method introduced errors (Supplementary Note 2). In particular, up to 20% of falsely called variants are not assigned any genome context features, so StratoMod uses only depth of coverage and variant allele fraction for classification. This limits StratoMod's ability to be as comprehensive as expected from a variant caller (Supplementary Fig. 15). Adding slop (i.e., extra bases around the repeat) can be important for some features, and we experimented with adding 1–5 bp slop around homopolymers. We decided to use 1 bp to simplify interpretation around nearby homopolymers, but this increased the number of variants without a genome context feature,

particularly for GC-rich regions. Despite this lack of comprehensiveness, the model's interpretability enables a more detailed understanding of how types of repeats and their characteristics predict sequencing and mapping error rates for PCR-free vs. PCR-plus Illumina sequencing. Understanding differences in sequencing and mapping errors between technologies can be challenging due to differences in how candidate variants are generated, so we did not use the model to predict false positives across technologies but instead predicted differences in false negatives after filtering for DeepVariant-based callsets from Illumina and HiFi in the second use case.

StratoMod is available as an open-source snakemake pipeline for anyone to use (see "Data availability" section). End users will derive the most benefit by training StratoMod themselves on their specific applications (modeled on the demonstrations provided here) as it is impossible for us to cover every combination of variant caller, mapper, platform, library prep, etc here. The memory requirements will vary widely depending on the number of VCFs one wishes to study at once, but in general, will require at least 12 G minimum (32 G is more realistic for more than two VCFs). The EBM training step is the main CPU-intensive task and will finish in about 60 min with 4 threads running on a 3 GHz Xeon processor when analyzing 2 VCFs with 22 features (note EBMs are not GPU-accelerated so this is not a consideration).

Users should also note that StratoMod was designed to work with germline variant calls, and will likely work best on whole genome sequencing (WGS) datasets. Other types of data such as whole exome sequencing and targeted sequencing may also work but will face the issue of having relatively fewer variants for training, as well as needing additional features to account for coverage variability and challenges near the edges of targeted regions. StratoMod additionally was not designed to assess somatic variant calling pipelines (although in theory, this may work with the right training data). Furthermore, the ClinVar callset we used likely has a small fraction (2%) of somatic variants in it[29]; however we do not believe this is a large enough number to significantly change our conclusions in Fig. 3. Furthermore, the ClinVar variants used in the gnomAD comparison (Fig. 5) were entirely germline since we only used variants common to both, and gnomAD is a database of germline variants.

StratoMod can be used for either explainability or prediction (which may involve explainability); either use case has different data requirements. Both require at least one benchmark VCF and at least one query VCF for training. In the former case, this is all that is needed, since the goal is to use the model to understand how different genome contexts lead to a given benchmarking outcome (analogous to finding the slopes in linear regression). In the latter case, one is concerned about predicting (and possibly explaining) the outcome of a variant outside the training set. Importantly, these predictions are conditional on priors (Supplementary Note 4) which for recall, precision, and Jaccard index are the likelihood of a variant being in the benchmark, the query, and either the benchmark or query respectively. In order for this to be useful, one needs to use past experience, data, and intuition regarding the likelihood of these priors.

The work presented here supports future stratification development in a more data-driven way. More comprehensive benchmarks, such as those based on de novo assembly, will also provide more accurate models of variant call errors, particularly in more difficult regions and for larger variants. We also expect these models along with manual curation to help systematize the creation of new benchmarks by helping to understand tradeoffs for each technology and variant caller so that we can know which method to trust when they differ. Models like StratoMod will provide a new approach both for developing better benchmarks and for using these benchmarks to understand strengths and weaknesses of a method and predict which clinically relevant variants may be missed.

## Methods
### Variant labeling
The overall process to label a query VCF file such that it can be understood by the EBM model is given in Fig. 1b. The following is a more detailed description of this process:

**Preprocessing**. To treat DeepVariant's filtered variants as variants when doing the comparison with vcfeval, we converted genotypes from ./. to 0/1 and 0/0 to 0/1. Furthermore, we removed all chromosomes except 1–22 (as the benchmarks we used did not have X/Y or alternate chromosomes). We also split multiallalic variants using bcftools norm --multiallelics -. Finally, we removed the MHC region from the benchmark and query VCFs before comparison.

**Comparison**. We generated TP, FP, and FN labels using vcfeval with '--refoverlap –all-records' to preserve all filtered variants (which were either kept or removed depending on the desired analysis). The output vcf files corresponding to TP, FP, and FN labels were then concatenated and converted to a bed file with an additional column holding the corresponding label for each variant. During this step, we also computed all VCF_* features (see below) from the VCF file itself. We also removed variants whose REF and ALT were both >1 bp and equal to each other and structural variants (those whose REF or ALT columns were longer than 50 bp). Importantly, we shifted the start/end columns resulting from the VCF file leftwise by 1 to make the final result 0-based instead of 1-based for proper intersection with BED files.

The truth set for each comparison was either the GIAB v4.2.1 benchmark for the indicated genome[2] or the Q100 draft assembly-based benchmark in the case of HG002 where noted. The HG002 Q100 draft small variant benchmark was created using v0.011 of DeFrABB (https://github.com/usnistgov/giab-defrabb), the T2T-HG002-Q100v1.0 diploid assembly (https://github.com/marbl/hg002), and GRCh38 reference (https://ftp-trace.ncbi.nlm.nih.gov/ReferenceSamples/giab/data/AshkenazimTrio/analysis/NIST_HG002_DraftBenchmark_defrabbV0.011-20230725/). DeFrABB (Development Framework for Assembly-Based Benchmarks) is a snakemake based pipeline created to facilitate the iterative development of benchmarks sets for evaluating variant callsets using high-quality diploid assemblies. DeFrABB first generates assembly-based variant calls using dipcall v0.3 (https://github.com/lh3/dipcall)[30]. Dipcall was run with default parameters with the following Z-drop parameter, -z200000,10000,200, which yielded more contiguous assembly-assembly alignments compared to the default value. After reformatting and annotation the variants reported by dipcall (vcf) are used as the draft benchmark variants. The benchmark regions are defined as regions with a 1:1 alignment between each assembled haplotype and the reference (except X&Y) and then excluded gaps in the assembly and their flanking sequences, as well as any large repeats (satellites, tandem repeats >10 kb, and segdups) that have a break in the assembly to reference alignment on either haplotype. Additionally structural variants including repeat regions when SVs overlapping large tandem repeats are also excluded from the benchmark regions. Widened SV coordinates were identified using the SVanalyzer v0.36 widen module (https://github.com/nhansen/SVanalyzer).

**Annotation**. After generating all feature bed files (see below section) we merged the SNV and INDEL label bed files from the previous step using multiple rounds of bedtools intersected with the -loj flag.

**Pre-train processing**. Prior to use in the EBM for training, we converted the labels (TP, FP, FN) to a 0/1 feature as required for binary classification. For the precision model, we simply removed the FN label and mapped FP -> 0 and TP -> 1. For the recall models, we wanted to remove the effect of including filtered variants in the training set (as if we had never used '--all-records' with vcfeval), and thus for all non-PASS variants, we mapped FP -> TN and TP -> FN, filtered only FN and TP labels, and then mapped TP -> 1 and FN -> 0. For the overlap model, we used the same mapping as the recall model except all PASS FP variants were also mapped to 0.

Since we used the -loj flag in the previous annotation step, many variants did not have any values for a given feature (since on average a given variant will only intersect with a few regions corresponding to our feature categories). For HOMOPOL_imperfect_frac and TR_percent_AT_median

we filled these missing values with −1 since 0 had real meaning for these features. For all other features we filled in missing values with 0; for these features, this corresponded to 0-length or 0-count, and thus made numerical sense as a missing substitute.

### Feature engineering
Model features were grouped into 5 main categories. See Supplementary Table 1 for a list of all features as used throughout the models in this work. The prefix of each feature corresponds to its category.

Note that VCF_DP and VCF_VAF were used only for the PCR-free/plus precision prediction model (Supplementary Note 1 and Supplementary Fig. 5) and none of the models in the main text. All other features were used in all models.

An overview of each feature category and their method of generation follows:

**VCF features (prefix = VCF).** VCF_DP and VCF_VAF were taken from the query VCF file without modification. VCF_input was used as an index to track the query VCF file where multiple inputs were used in the model (which in this work was used to represent different mappers, different sequencing technologies, or different library preps). VCF_indel length was taken as the difference between the ALT and REF columns (thus positive values represented insertions).

**Homopolymers (prefix = HOMOPOL).** Perfect homopolymers (e.g. homopolymers with no other interrupting bases) with lengths ≥4 bp were generated directly from the reference using an in-house Python script and saved to a bed file. This bed file was then split into each of the four bases. Each individual homopolymer-per-base file was then merged using bedtools with -d 1 (to get "imperfect" homopolymers which have at least two stretches of the same base ≥4 bp with one different base in between). We then added 1 bp slop to each end of the merged regions in order to detect errors immediately adjacent to the homopolymer itself. For each homopolymer region, we used the length (without slop) as well as the fraction of imperfectness (the number of non-homopolymer bases over the length). Note that in our formulation, imperfect fraction approaches a theoretical maximum of 20% in the limit as length increases.

**Tandem Repeats (prefix = TR).** Tandem repeat features were based off of the simple repeat finder UCSC database. We used bedtools merge to summarize a subset of the columns in this database (period, copyNum, perMatch, perIndel, and score; note that we renamed these in our feature set to better distinguish them). For each of these columns, we computed the min, max, and median values when merging, and also stored the count of the number of merged repeats. Furthermore, we computed the percent of GC and AT content in each region using the individual base percent columns present in the database and merging analogously to the previous columns. Finally, computed the length of the tandem repeat region directly using the coordinates present in the database file. We added 5 bp slop to each region and removed all regions with period/unitsize == 1 as these corresponded to homopolymers which were represented in a different feature category.

**Segmental duplications (prefix = SEGDUP).** Segmental duplication features were merged in a similar manner to the tandem repeat columns, except we used the genomicSuperDups database from UCSC. We only used the alignL and fracMatchIndel columns and computed the min, max, and mean of these as well as the count of regions that were merged. We did not add slop to these features.

**Repeat Masker (prefix = REPMASK).** Repeat masker features were based on the rmsk database from UCSC. For entries whose class was SINE, LINE, or LTR, we filtered by class and merged using bedtools. We then calculated the length of each merged region (conditioned on class).

We did not add slop to these regions. In the case of SINE and LTR, we converted each feature to binary by setting each length to 1 (and then any non-intersecting variants would get a 0 representing that they did not intersect this region).

**Hard-to-map (prefix = MAP).** We used the GEM-based[11] low-mappabilityall.bed.gz (100 bp) and nonunique_l250_m0_e0.bed.gz (250 bp) from the GIAB v3.0 stratifications[1] as the basis for this feature. For both of these bed files, we simply appended a column filled entirely with 1's representing a binary feature where 1 means "in a hard-to-map region" and 0 otherwise.

### Model training
We used the interpretml package[31] from Microsoft Research to train the EBM models. We specifically used the ExplainableBoostingClassifier which has the following form:

$$g(x) = f_1(x_1) + f_1(x_1) + \ldots + f_{1,2}(x_1, x_2) + \ldots$$

Here $g$ is the logit link function and each $f_i$ is either a univariate or bivariate decision tree in terms of its input(s).

These were trained using a random 80/20 train/test split. For interactions, we specifically included all interactions that included VCF_input as well as indel_length vs all four of the homopolymer length features. All other settings were left at default.

### Model comparisons
To compare EBMs with other commonly used models in the machine learning space, we used the FP/TP EBM model with Illumina PCR-Free/Plus and ran the associated data through the algorithms and hyperparameter tuning schemes described in Table 1.

### ClinVar analysis
We used the November 5 2022 release of the ClinVar VCF. We added a FORMAT (GT) and SAMPLE (0/1) column to the VCF and used hap.py (https://github.com/Illumina/hap.py) to compare the v4.2.1 benchmark VCF and BED (https://ftp-trace.ncbi.nlm.nih.gov/ReferenceSamples/giab/release/ChineseTrio/HG005_NA24631_son/NISTv4.2.1/GRCh38/). Restricting to "likely pathogenic" or "pathogenic" variants in the ClinVar VCF, this resulted in 133 matching variants from HG005.

### ClinVar gnomAD validation
The model was trained using our draft Q100 benchmark and a 30× Illumina-DeepVariant callset as the query (which we picked since we assumed it would closely match the data used to make gnomAD). The model was trained as described above using both FP and FN as the negative

**Table 1 | Parameters used for models when comparing to EBMs**

| Model | Implementation | Hyperparameter levels |
|---|---|---|
| Decision tree | rpart (R) | Cost_complexity: 0.00001, 0.0001, 0.001, 0.01, 0.1 |
| Logistic regression | glmnet (R) | Penalty: 0.000001, 0.000001, 0.00001, 0.0001, 0.001, 0.01, 0.1, 1, 10<br>Mixture: 0, 0.5, 1 |
| Random forest | ranger (R) | mtry : 1, 4, 7<br>trees: 500, 1000, 2000 |
| XGBoost | xbgoost (python/gpu accel) | max_depth : 3, 6, 9<br>n_estimators: 100, 500, 1000<br>gamma: 1, 10, 100 |

All models (including the EBMs) were trained on a compute cluster with 512 GB memory, 2 20-core Intel Xeon E52698 v4 CPUs, and 8 Nvidia Tesla V100 (per node). Each job was allowed 3 days of compute time. Of all the algorithms used (including EBMs), only xgboost was able to take advantage of GPU acceleration.

class and with FILTERed variants removed in the benchmark comparison. The ClinVar VCF (described above) was the input into this trained model to obtain prediction probabilities for either TP (no error) or FP/FN (error).

The ClinVar VCF (modified now with model probabilities in the FORMAT column), was then intersected with the genomic gnomAD v4.0 vcf (chromosomes 1–22 since this was what was included in our model) using bcftools isec with default parameters.

The gnomAD VCFs were individually downloaded from https://gnomad-public-us-east-1.s3.amazonaws.com/release/4.0/vcf/genomes/gnomad.genomes.v4.0.sites.chr%i.vcf.bgz where *i* is 1 through 22.

## Statistics and reproducibility
The StratoMod pipeline uses snakemake as its base. Reproducibility is ensured by specifying all inputs as publicly-accessible URLs and hashing the downloaded contents to alert the user when the source changes and potentially may change the output. This configuration is then committed to git. Additionally, the pipeline fully specifies each of its conda environments (including version and build) for both individual rules and overall runtime.

Code integrity is ensured using a continuous integration/continuous development pipeline (run on a privately-hosted GitLab) which includes linting with mypy/pylint as well as snakemake integration tests using small-scale datasets. See the link to the repository below for more details.

See Supplementary Table 5 for a list of software packages and their versions that were used in this work.

## Reporting summary
Further information on research design is available in the Nature Portfolio Reporting Summary linked to this article.

## Data availability
All data can be found on figshare at https://figshare.com/account/home#/projects/164446.

## Code availability
The pipeline repository is located at https://github.com/usnistgov/giab-stratomod[32]. Experiments using this pipeline are located at https://github.com/usnistgov/giab-stratomod-experiments[33]. Details on each of the datasets used can be found in Supplementary Table 6[34–48].

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

## Acknowledgements
We thank Jennifer McDaniel for developing many of the stratifications used as a basis for this work. Certain commercial equipment, instruments, or materials are identified to specify adequately experimental conditions or reported results. Such identification does not imply recommendation or endorsement by the National Institute of Standards and Technology, nor does it imply that the equipment, instruments, or materials identified are necessarily the best available for the purpose.

## Author contributions
N.D. developed the StratoMod pipeline; N.D. and J.W. ran experiments and generated figures; N.D., P.T., N.O., F.S., J.W., and J.Z. wrote the manuscript.

## Competing interests
F.J.S. has received support from Oxford Nanopore Technologies, Pacific Biosciences, Illumina, and Genentech. None of the other authors declare conflicts of interest.
