## [Transparent Peer Review file · Communications Biology]

StratoMod: Predicting sequencing and variant calling errors with interpretable machine learning

Corresponding Author: Dr Nathan Dwarshuis

A version of this paper was originally rejected for publication by Communications Biology, however that decision was reconsidered after appeal by the authors.

Version 1:

Reviewer comments:

Reviewer #2

(Remarks to the Author)

The authors present StratoMod, a computational method for predicting whether a given variant would be correctly detected (a true positive) or not detected (a false negative) for a specific sequencing technology and variant calling pipeline. The method uses an easily interpreted machine learning model, Explainable Boosting Machines (EBMs), and training data from Genome in a Bottle consortium materials. By inspecting the parameters ("feature plots") of the EBMs, the genomic features that cause difficulties for a specific sequencing technology and computational pipeline can be identified. It may also be possible to use the method to compare different technologies/pipelines at subsets of sites of interest (e.g., pathogenic variants for a given disease) to determine the best variant calling approach for a given application (although see comment #2).

Overall, the presented approach allows for a rigorous evaluation of genomic features that create challenges for variant calling approaches and is likely useful for developers of such approaches. However, it is unclear how useful the method is to users of variant calling approaches, and thus the scope of its immediate impact is limited. In addition, there are a number of aspects of the results and experimental methodology that are hard to follow.

Major comments:

1. It needs to be made explicitly clear what the prediction of a "false negative" by StratoMod means. If I am understanding correctly that the model is trained using only known (benchmark) variant sites which are labeled either true positive (TP) or false negative (FN), then it follows that the predictions are *conditional on their being a true variant at the site*. This is very different from a probability of a given site being a false negative regardless of knowing, a priori, that there is truly a variant at that site. Phrases in the abstract ("a new method of finding likely false negatives") and elsewhere in the text are misleading in that they suggest the latter (unconditional) type of probability, which would indeed be an exciting development, but is not what is described.
2. Given that the StratoMod probabilities are conditional, they are most useful *prior* to performing a sequencing and variant calling experiment. For example, given a subset of potential variants of interest, a researcher could use StratoMod to determine which sequencing technology and computational pipeline would be best to detect those variants, should they be present. However, two of the features that StratoMod uses as input are read depth and allele frequency at a given site, which would only be obtained after having done the experiment. I recommend that the authors clarify how they expect StratoMod to be used in this case, if this is indeed an use case that is targeted.
3. I did not understand a number of aspects of the ClinVar experiments:
 - 3A. First, the presented model uses read depth and allele frequency from the VCF as features. However, it is not clear to me that a VCF from ClinVar would have such information, as it is not specific to a given sequencing experiment (as far as I understand).
 - 3B. Second, the text says that the ClinVar VCF was used "without labels." It is not clear what that means here, since the model is being used to predict.

3C. Why are TP probabilities bisected at 90%? If the model probabilities are well calibrated then any site with TP probability > 50% is more likely to be a TP and any site with TP probability < 50% is more likely to be a TN (again, conditional on there actually being a true variant at this site).

4. I did not understand the section "Performance assessment and prediction of missing clinically-relevant variants" - how is this different from what was done in the section "Identifying driving features that predict False Negative Variant Calls"? The experimental methodology sounds the same.

5. allele count (AC) was shown to be an important feature - why not include this as a feature in Stratomod?

Minor comments:

6. Figure 6A - For SNVs, the red curve is labeled as for allele count < 10, but it closely matches the red dotted line (agreement between predicted probability and PASS fraction). The blue line (allele count > 10) is lower, corresponding to Stratomod being overconfident for those variants. This seems to contradict the text. Were the colors accidentally switched?

7. Figure 3D - Please provide informative labels for the various tracks - they are largely uninterpretable

8. Page 11 - "VG was often more confident than BWA" - I think this language is misleading/incorrect. If I understand correctly, "confidence" here refers to the Stratomod's predicted probability of TP *not* a confidence score assigned by a variant caller using VG.

9. Page 21 - "VCF_input was used as an index to track the query VCF file where multiple inputs were used in the model." I did not understand this sentence until I read the supplementary material. Please clarify the purpose of the VCF_input feature in the main text.

10. Page 21 - ">= 4pb" -> ">= 4bp"

Reviewer #3

(Remarks to the Author)

Dwarshuis et. al present a novel deep learning approach called StratoMod that classifies genomic regions according to their difficulty for valid variant calling. Thereby, StratoMod provides a user with information on how likely it is to observe false positive or false negative results and thus, eases the decision on whether to move to a different sequencing technology or not.

While StratoMod presents an interesting approach, considering the important but often unaddressed topic of false negative variant calls, I have several concerns:

Major:

* MNVs are removed – why? It is relatively easy to break down MNVs to a set of SNVs and Indels. As MNVs play an important role in variant calling, they should not be excluded from analysis. Furthermore, what is considered the threshold between Indels and SVs? To remove translocations and inversions seems understandable, especially when also considering short-read sequencing. But for insertions and deletions – at which number of base pairs are they considered an SV and thus removed? Are different thresholds applied for short- and long-read seq data?

* If I understand correctly, the authors just used DeepVariant for variant calling. Why? The caller is far from being the common standard in variant calling. GATK, MuTect, SAMtools, VarScan, FreeBayes etc. are much more established and used much more often. Furthermore, it seems to me that the selected variant caller will have a major influence on the model training and also on the results. Thus, the authors should apply several common variant callers, report their results and analyse their differences. I would suggest 3-5 different tools.

* I am not sure I understood the section about false negatives in the ClinVar data correctly. Do the authors predict the probability to miss a variant known from ClinVar? However, ClinVar is a continuously growing database. What is in there, was apparently not missed. But what is not yet in there – wouldn't those be the variants we expect to have missed so far? Furthermore, many pathogenic variants in ClinVar are related to cancer, which means that they might be present with very low allele frequencies (sub-clonal mutations). They are a challenge on their own. Deep sequencing data, commonly targeted seq would be necessary to detect them. DeepVariant is certainly not the right tool to detect these calls. Furthermore, I could not find VCF_VAF in Supp figure 1. This is very surprising.

* To me, unfortunately, it does not become clear how I should exactly use StratoMod. In case I have sequencing data, targeted seq from Illumina NextSeq and variant calls with GATK. Can I directly use StratoMod to distinguish TP, FP and FN calls? Or is new training/fine-tuning required? And what if I switch to Illumina NovaSeq and FreeBayes? Or from targeted seq to WES? Or switching the disease – saying from cancer to male infertility, which both have genetic components. The authors should work out the application of StratoMod to novel data sets and its benefit more clearly.

Minor:

* The authors should carefully take care of a uniform spelling of their approach – StratoMod vs Stratomod. Same is true for, e.g. Clinvar vs ClinVar.

* In the figures, the different sections are commonly labelled as A., B. etc. But in the legend it is a) and b) for Figures 1, 2, 4 and 6, A), B) etc. for Figure 3.

* Figure 1: Abbreviations like LINEs and SINEs should be introduced. Part a) is not thoroughly clear to me. How does the information on the top fit together with the blue boxes below? Should they match the 4 steps? Can we actually see quantification of the region in the figure?

* Figure 5 is missing – maybe a wrong label of Figure 6?

* The resolution of the IGV screenshots should be improved.

* The authors mention 22 features, but in Supp table 1 I only find 17? The categorical feature (seq platform) is not included. Regarding the other 4 – I don't know what they are.

* A reference for BWA is missing. Do the authors use BWA mem? Same for vg (which is not spelled VG).

* As the authors use gnomAD 4.0, which was released in Nov 2023, they should not use a ClinVar release from Nov 2022.

* When trying out StratoMod: The exemplary file is not called dynamic-testing.yml, but testing.yml. The authors should update this information in their git-repository. Unfortunately, I could not successfully execute the example, the error message was: `pip._vendor.urllib3.exceptions.ReadTimeoutError: HTTPSConnectionPool(host='files.pythonhosted.org', port=443): Read timed out. failed CondaEnvException: Pip failed`

Version 2:

Reviewer comments:

Reviewer #2

(Remarks to the Author)

The authors have satisfactorily addressed my prior comments. I particularly appreciate the changes in phrasing that make it more clear what StratoMod is predicting. I have a few minor follow-up comments on the revised manuscript:

1. Figure 1c: I understand that one could predict the Jaccard Index, but I have hard time seeing any practical use of this and thus I suggest removing this scenario from the figure as it leads to confusion. In particular, to compute the jaccard index, one needs to predict at every position that is either in the benchmark or in the query - but if you know the positions in the benchmark and query you can directly compute the jaccard index without any prediction. I don't see a scenario in which you know that a position is in either the benchmark or query but do not know if it is in both.

2. Now that I better understand the section "Performance assessment and prediction of missing clinically-relevant variants", I believe there is still some confusing language in this section with regard to "precision" and "recall". In much of the rest of the manuscript, "precision" and "recall" refer to the variant calls of a particular *sequencing pipeline*. However, in this section, if I understand correctly, "precision" and "recall" refer to the predictions of *StratoMod* at ClinVar variant positions. Some clarification of this language would help the reader.

Reviewer #3

(Remarks to the Author)

Most of my comments were addressed. I just have one more remark: The authors commented on ClinVar containing only germline variants until 2024. This is not thoroughly correct. It is true that ClinVar recently introduced "an enhanced data model to better accept and support classifications of somatic variants". But: it contained somatic variants even before that. For example NRAS G12A (<https://www.ncbi.nlm.nih.gov/clinvar/variation/219097/>). It is in there since 2016 and a very common cancer variant. I have seen this variant many times in various data sets. It is a somatic variant, often present at low VAFs. According to a paper by Ritter et al. (Genome Medicine 2016), ClinVar contained - at that time - around 2% somatic variants. Maybe the authors could comment on this? As it affects their approach, especially the use of DeepVariant and the focus on not-WES and not-targeted seq data.

Version 3:

Reviewer comments:

Reviewer #2

(Remarks to the Author)

In this revision, the authors have added further clarification to a couple of confusing aspects of the manuscript. Although I continue to find the highlighting of the Jaccard index prediction to be confusing and of limited practical use (and thus likely better suited to the supplementary materials), I do not object to its inclusion in the main text.

Dear reviewers,

Thank you for taking the time to review our manuscript entitled “StratoMod: Predicting sequencing and variant calling errors with interpretable machine learning”. Please see the following for our responses to your feedback.

Note that all line numbers refer to the main text. Our responses are in blue, occasionally referring to specific feedback in red.

Reviewer #2:

The authors present StratoMod, a computational method for predicting whether a given variant would be correctly detected (a true positive) or not detected (a false negative) for a specific sequencing technology and variant calling pipeline. The method uses an easily interpreted machine learning model, Explainable Boosting Machines (EBMs), and training data from Genome in a Bottle consortium materials. By inspecting the parameters ("feature plots") of the EBMs, the genomic features that cause difficulties for a specific sequencing technology and computational pipeline can be identified. It may also be possible to use the method to compare different technologies/pipelines at subsets of sites of interest (e.g., pathogenic variants for a given disease) to determine the best variant calling approach for a given application (although see comment #2).

Overall, the presented approach allows for a rigorous evaluation of genomic features that create challenges for variant calling approaches and is likely useful for developers of such approaches. However, it is unclear how useful the method is to users of variant calling approaches, and thus the scope of its immediate impact is limited. In addition, there are a number of aspects of the results and experimental methodology that are hard to follow.

Major comments:

1. It needs to be made explicitly clear what the prediction of a "false negative" by StratoMod means. If I am understanding correctly that the model is trained using only known (benchmark) variant sites which are labeled either true positive (TP) or false negative (FN), then it follows that the predictions are *conditional on their being a true variant at the site*. This is very different from a probability of a given site being a false negative regardless of knowing, a priori, that there is truly a variant at that site. Phrases in the abstract ("a new method of finding likely false negatives") and elsewhere in the text are misleading in that they suggest the latter (unconditional) type of probability, which would indeed be an exciting development, but is not what is described.

This was an excellent point and led us to clarify how StratoMod should be interpreted.

It is indeed true that what StratoMod is predicting is conditional as a consequence of subsetting to a given error label (FP, FN, or both). To be more precise (for the case of TP vs FN), the probability being predicted by StratoMod is $P(Q|G)$ where Q = in query and G = in genome/benchmark (benchmark being a genome for which the truth is known and thus a model is trainable). The "probability of a false negative" in the variant calling context is $P(!Q \text{ and } G)$ (variant not in query but in genome/benchmark), which we assume is what was meant by "unconditional false negative." This is indeed different from what StratoMod is predicting, since $P(!Q|G) \neq P(!Q \text{ and } G)$, and thus saying "false negative" in the manner we did was misleading.

To address this, we significantly modified Figure 1 (specifically added panel c) to show how subsetting to different variant calling results leads to different interpretations.

a.

b.

c.

Particularly, subsetting to TP and FN can be interpreted as “predicted recall” (since the definition of recall as $TP/(FN+TP)$ directly corresponds to the conditional probability $p(Q|G)$ as described above). We made similar distinctions for precision and jaccard index which use $FP+TP$ and $FP/FN+TP$ respectively, since we use all three interpretations throughout the manuscript. We also reflected these interpretations throughout the text and latter figures by substituting “ $p(TP)$ ” for “predicted recall/precision/overlap” (depending on the model). We also replaced all mentions of TP and FN with “detected variant” and “missed variant” in the case of “predicted recall” models to reflect that the benchmark is in the denominator. We similarly replaced TP and FP in the case of the predicted precision models (Supplemental Note 2) with “correctly called variant”

and “incorrectly called variant” which reflects that the query is in the denominator. This had the added benefit of helping us distinguish between the interpretation of StratoMod and its performance in the text (how well the model is doing its classification, which is also assessed by counting TP/FP/FN/TNs but has nothing to do with the labels of each variant call).

However, we respectfully disagree that this is uninteresting to the genomics community. For example, clinical labs typically need to report sensitivity for variants (sometimes called “recall” when benchmarking or “positive predictive agreement” in the FDA guidance about technical accuracy

<https://www.fda.gov/regulatory-information/search-fda-guidance-documents/considerations-design-development-and-analytical-validation-next-generation-sequencing-ngs-based>), and

Stratomod is predicting the recall of a method for each variant in ClinVar. StratoMod can also be used to predict the precision in a similar way, for which we provide an example (Supp Note 2) but focus on recall here since we are not aware of any other methods that predict missed variants and we believe this would be especially valuable for clinicians (because they can curate potential incorrectly called variants but would be blind to false negatives). Furthermore, performance of variant calling pipelines is often measured using precision and recall, which is exactly what StratoMod is predicting.

Furthermore, if one is interested in obtaining the “unconditional probability of a false negative,” this can easily be found using StratoMod’s recall prediction if one also knows the frequency of the variant in question. More precisely, if we know $P(!Q|G)$ from StratoMod and we also know $P(G)$ (the likelihood the variant is in the genome), then it follows by the chain rule of probability that $P(!Q \text{ and } G) = P(!Q|G) * P(G)$. $P(G)$ can be obtained from gnomAD or any other population-level variant database. We added this point to the discussion (see line 648) since we had not considered this use-case previously, and elaborate on the math more thoroughly in Supp Note 4 (which was also added).

2. Given that the StratoMod probabilities are conditional, they are most useful *prior* to performing a sequencing and variant calling experiment. For example, given a subset of potential variants of interest, a researcher could use StratoMod to determine which sequencing technology and computational pipeline would be best to detect those variants, should they be present. However, two of the features that StratoMod uses as input are read depth and allele frequency at a given site, which would only be obtained after having done the experiment. I recommend that the authors clarify how they expect StratoMod to be used in this case, if this is indeed an use case that is targeted.

We only used VAF and DP in one model, which was the PCR-free/plus comparison in Supplemental Note 1/Supplemental Figure 5 where we predicted precision instead of recall. Supplemental Table 1 also showed that these two features were only used in the aforementioned model. However, this point was not made in the Methods section where we talk about DP/VAF so this has been fixed (line 941).

While it is true that StratoMod's most obvious use-case is to predict the result of a pipeline a priori, in the PCR-free/plus experiment we also used StratoMod's interpretable features to understand the behavior of the two library preparations with respect to incorrectly calling variants. For example, we used the segmental duplications feature to trace back incorrect calls to a specific region in GRCh38 in the TCR locus. Thus StratoMod can be used to troubleshoot a pipeline after it is run in addition to raw prediction.

3. I did not understand a number of aspects of the ClinVar experiments:

3A. First, the presented model uses read depth and allele frequency from the VCF as features. However, it is not clear to me that a VCF from ClinVar would have such information, as it is not specific to a given sequencing experiment (as far as I understand).

We actually did not include these (with the exception of one model), but the methods section did not make this clear. This has been fixed (line 941, "Note that VCF_DP and VCF_VAF were used only for the PCR-free/plus precision prediction model")

3B. Second, the text says that the ClinVar VCF was used "without labels." It is not clear what that means here, since the model is being used to predict.

This was redundant and thus misleading, so we removed this phrase (line 283). This was meant to emphasize that there is no benchmark for Clinvar and thus we didn't try to test the Clinvar VCF against ground truth.

3C. Why are TP probabilities bisected at 90%? If the model probabilities are well calibrated then any site with TP probability > 50% is more likely to be a TP and any site with TP probability < 50% is more likely to be a TN (again, conditional on there actually being a true variant at this site).

While the 90% cut-off is somewhat arbitrary, we think it is reasonable for users of a method to know ClinVar variants that they have a >10% probability of missing if they are present in a sample, which could be important (i.e., the model predicts their sensitivity or recall would be <90%). We also assume the reviewer may have had a typo for "TN" above, because any site with TP probability < 50% is more likely to be a FN, which would also be a reasonable cut-off but would include fewer variants.

In reality, the true cutoff needs to be derived from a risk assessment that only the end user can perform. We clarified this point in the discussion (paragraph starting at line 642)

4. I did not understand the section "Performance assessment and prediction of missing clinically-relevant variants" - how is this different from what was done in the section "Identifying driving features that predict False Negative Variant Calls"? The experimental methodology sounds the same.

That's probably because we had an extra paragraph in the latter section you mentioned that we neglected to delete after moving it to its own section. This has been fixed (line 201). We also moved this performance section higher up since this made the story flow better (even though in our estimation performance is the least interesting aspect of this work) (line 260 and 357).

5. Allele count (AC) was shown to be an important feature - why not include this as a feature in Stratomod?

While this might make sense for certain applications, adding allele count to the training process would be quite tricky and produce many biases that would be unhelpful. Since we used the long read assembly-based Q100 benchmark for most of our analysis, many challenging variants are not in the short read-based gnomAD and would not have an allele count. We could circumvent this by setting these to 0 (or taking them out, which defeats the purpose of using Q100). If we set these to 0, many of these missing variants would correlate to difficult regions, which would confound with many of our other features (segmental duplications, hard to map, etc) and in this way it would likely not add information and possibly make the model worse.

Minor comments:

6. Figure 6A - For SNVs, the red curve is labeled as for allele count < 10, but it closely matches the red dotted line (agreement between predicted probability and PASS fraction). The blue line (allele count > 10) is lower, corresponding to Stratomod being overconfident for those variants. This seems to contradict the text. Were the colors accidentally switched?

The colors are correct, but this is a complex point so we have now substantially revised this results section. In particular, the confusion was likely related to most of the variants being in the large point in the upper right of the plots, so we have now specifically highlighted this in the text (paragraph starting at line 562). We have also changed some of the terminology in this section to be more explicit about what Stratomod is predicting (error rate) and what gnomAD is doing (filtering candidate calls).

7. Figure 3D - Please provide informative labels for the various tracks - they are largely uninterpretable

We added bigger text to fix this.

a.

b.

c.

d.

8. Page 11 - "VG was often more confident than BWA" - I think this language is misleading/incorrect. If I understand correctly, "confidence" here refers to the Stratomod's predicted probability of TP *not* a confidence score assigned by a variant caller using VG.

Fair point, this has been clarified (paragraph starting at line 400).

9. Page 21 - "VCF_input was used as an index to track the query VCF file where multiple inputs were used in the model." I did not understand this sentence until I read the supplementary material. Please clarify the purpose of the VCF_input feature in the main text.

This has been fixed (line 951, "which in this work was used to represent different mappers, different sequencing technologies, or different library preps")

10. Page 21 - ">= 4pb" -> ">= 4bp"

This has been fixed (line 961).

Reviewer #3:

Dwarshuis et. al present a novel deep learning approach called StratoMod that classifies genomic regions according to their difficulty for valid variant calling. Thereby, StratoMod provides a user with information on how likely it is to observe false positive or false negative results and thus, eases the decision on whether to move to a different sequencing technology or not. While StratoMod presents an interesting approach, considering the important but often unaddressed topic of false negative variant calls, I have several concerns:

Major:

1. MNVs are removed – why? It is relatively easily to break down MNVs to a set of SNVs and Indels. As MNVs play an important role in variant calling, they should not be excluded from analysis. Furthermore, what is considered the threshold between Indels and SVs? To remove translocations and inversions seems understandable, especially when also considering short-read sequencing. But for insertions and deletions – at which number of base pairs are they considered an SV and thus removed? Are different thresholds applied for short- and long-read seq data?

MNVs are removed – why?

This was a typo from an earlier version of StratoMod where we did remove MNVs; in the current version we split them as you implied (clarified in lines 167 and 873).

What is considered the threshold between Indels and SVs?

SVs are INDELS > 50 bp. We explained this in the methods and added a clarifying remark to the results (line 167). This applies to both long and short reads, and is a commonly accepted (albeit somewhat arbitrary) convention used in the sequencing development/variant calling community.

2. If I understand correctly, the authors just used DeepVariant for variant calling. Why? The caller is far from being the common standard in variant calling. GATK, MuTect, SAMtools, VarScan, FreeBayes etc. are much more established and used much more often. Furthermore, it seems to me that the selected variant caller will have a major influence on the model training and also on the results. Thus, the authors should apply several common variant callers, report their results and analyse their differences. I would suggest 3-5 different tools.

If I understand correctly, the authors just used DeepVariant for variant calling. Why? The caller is far from being the common standard in variant calling. GATK, MuTect, SAMtools, VarScan, FreeBayes etc. are much more established and used much more often.

StratoMod was designed for germline variant calling pipeline. We did not make this clear so we clarified this point in the abstract and elsewhere (see comment 3). The variant callers you listed (with the major exception being GATK) are indeed popular albeit in the somatic variant calling

context. GATK may be older and more mature than DeepVariant; however DeepVariant has achieved impressive enough results to be useful. For our purposes, DeepVariant enabled us to train models with a consistent variant caller across different sequencing technologies, which is a strength of DeepVariant.

Furthermore, it seems to me that the selected variant caller will have a major influence on the model training and also on the results. Thus, the authors should apply several common variant callers, report their results and analyse their differences. I would suggest 3-5 different tools.

It is true that the variant caller would have an impact on performance. However, StratoMod is designed to assess pipelines, of which variant callers are only one component. We demonstrated the abilities of StratoMod to model differences in errors between the library prep, platform, and mapper components of the pipeline. While we did not explicitly test any variant callers head-to-head, this is certainly another important component that StratoMod could model. Because variant calling methods evolve particularly rapidly, we chose not to use StratoMod to demonstrate which variant caller is “the best” (which will almost certainly change by the time the paper is read), but rather to make a tool to measure and understand pipeline performance.

That said, to show StratoMod is useful for assessing variant callers, we have added two callsets from Oxford Nanopore, for which the variable being tested was different iterations of the base/variant caller (see Supp Fig 5 and section starting at line 467).

a.

b.

3. I am not sure I understood the section about false negatives in the ClinVar data correctly. Do the authors predict the probability to miss a variant known from ClinVar? However, ClinVar is a continuously growing database. What is in there, was apparently not missed. But what is not yet in there – wouldn't those be the variants we expect to have missed so far? Furthermore, many pathogenic variants in ClinVar are related to cancer, which means that they might be present with very low allele frequencies (sub-clonal mutations). They are a challenge on their own. Deep sequencing data, commonly targeted seq would be necessary to detect them. DeepVariant is

certainly not the right tool to detect these calls. Furthermore, I could not find VCF_VAF in Supp figure 1. This is very surprising.

StratoMod is designed to be used on germline variants. We did not necessarily state that it could not be used on somatic variants, so we added some clarifying language in the abstract (line 19) and discussion (line 721) to deter this false assumption. This is likely part of the confusion.

Do the authors predict the probability to miss a variant known from ClinVar?

To address this question directly, we used StratoMod to predict which ClinVar variants are likely to be missed. Furthermore, we only predict pathogenic/likely pathogenic variants, which should limit the analysis to germline variants (since, according to the ClinVar documentation, pathogenic and likely pathogenic are germline by definition). Also, ClinVar officially started accepting somatic variants in May 2024, after we downloaded our ClinVar dataset. The target use-case for this analysis was a clinical laboratory needing to assess which pipeline to use when looking for pathogenic germline variants; as a demonstration we used long vs short reads since this seems like an obvious tradeoff one may need to make in terms of cost and accuracy (and indeed we show long reads are more accurate, and by how much).

However, ClinVar is a continuously growing database. What is in there, was apparently not missed. But what is not yet in there – wouldn't those be the variants we expect to have missed so far?

We understand ClinVar is continuously growing and more pathogenic variants will be added over time, particularly in the most challenging regions of the genome. Nevertheless, we show that many challenging variants are in ClinVar, particularly in certain genes in segmental duplications. As new challenging variants are added to ClinVar using new technologies like long reads, StratoMod will become increasingly useful for predicting variants that might be missed by other methods.

Furthermore, many pathogenic variants in ClinVar are related to cancer, which means that they might be present with very low allele frequencies (sub-clonal mutations). They are a challenge on their own. Deep sequencing data, commonly targeted seq would be necessary to detect them. DeepVariant is certainly not the right tool to detect these calls.

We have now clarified that the question of allele frequency (in the somatic context) is irrelevant since we are looking at germline variants. DeepVariant is also a common tool when assessing germline variants and thus is appropriate for this work, as explained above.

Furthermore, I could not find VCF_VAF in Supp figure 1. This is very surprising.

We did not use VCF_VAF in Supp Figure 1. The only model in which we used this (and VCF_DP) was in the PCR-free/plus model in Supp Note 1/Supp Fig5. Supp Table 1 shows this;

however the methods section was misleading in stating which models used which feature so we clarified this (see revised Supp Table 1 as well as line 941)

We included VAF and DP in this one model as a demonstration of how StratoMod can be used to troubleshoot a pipeline post-hoc. Since VAF and DP required a mapper and variant caller to have produced a VCF, training a model with this information will not help anyone predict how a given method will behave a priori (which was the point of all the models in the main text).

4. To me, unfortunately, it does not become clear how I should exactly use StratoMod. In case I have sequencing data, targeted seq from Illumina NextSeq and variant calls with GATK. Can I directly use StratoMod to distinguish TP, FP and FN calls? Or is new training/fine-tuning required? And what if I switch to Illumina NovaSeq and FreeBayes? Or from targeted seq to WES? Or switching the disease – saying from cancer to male infertility, which both have genetic components. The authors should work out the application of StratoMod to novel data sets and its benefit more clearly.

Can I directly use StratoMod to distinguish TP, FP and FN calls?

StratoMod can be used to distinguish TPs from FPs, and TPs from FNs, but not all three simultaneously, because the model is a binary classifier and thus can only predict two labels at once. One would either need to combine two of the labels to make a meta-label or do as we did and limit the analysis to one type of error. This has been clarified (see revised Figure 1).

a.

b.

c.

Or is new training/fine-tuning required?

We don't release the models because they are for particular pipelines, which become outdated quickly, so the user needs to train it themselves for their own pipeline(s) of interest. Sequencing companies are constantly revising their technologies, and likewise for the developers behind mappers and variant callers. We are also constantly updating StatoMod itself.

Therefore, the examples we posed in the manuscript are demonstrations of the tool's capabilities and not necessarily intended to reveal deep truths about various platforms, mappers, etc. Indeed, many of the results we found are generally known (such as long reads

having superior mapping to short reads). StratoMod allows these to be quantified and visualized in a precise way that was not previously possible.

Finally, the specifics of how to train/fine-tune new models are described in the Github repo so that they can be updated as StratoMod is refined.

This was clarified in the discussion as well (line 714, “End users will derive the most benefit by training StratoMod themselves...”).

And what if I switch to Illumina NovaSeq and FreeBayes?

While many of the general trends may be the same between our results for Illumina NovaSeq + DeepVariant and a different approach using FreeBayes, you would need to train a new model if you want to gain a precise understanding of how FreeBayes compares to DeepVariant (or if you’d want to compare Novaseq X to Novaseq 6000).

Or from targeted seq to WES?

We have not tested either of these so we don’t know for certain, However, both of these have included far fewer benchmark variants relative to whole genome sequencing (which we used). One could train StratoMod on targeted/WES directly, but it might not give the model enough variation to learn something useful.

Targeted/exome sequencing also has much higher variability in coverage, so it likely would be important to include additional features like coverage and whether the variant is near the edge of the targeted regions, so it is beyond the scope of this work.

We have clarified this limitation in the discussion (line 723, “Other types of data such as whole exome sequencing and targeted sequencing may also work, but will face the issue of having relatively fewer variants for training...”)

Or switching the disease

StratoMod is disease-agnostic. It is a tool designed to assess the tradeoffs between sequencing pipelines and their components, which could be used to study many different things.

However, see previous note about StratoMod being designed with germline variants in mind, and other previous note about this probably failing on targeted/WES.

Minor:

5. The authors should carefully take care of a uniform spelling of their approach – StratoMod vs Stratomod. Same is true for, e.g. Clinvar vs ClinVar.

This has been fixed throughout the text.

6. In the figures, the different sections are commonly labelled as A., B. etc. But in the legend it is a) and b) for Figures 1, 2, 4 and 6, A), B) etc. for Figure 3.

All panel letters are now lowercase.

7. Figure 1: Abbreviations like LINEs and SINEs should be introduced. Part a) is not thoroughly clear to me. How does the information on the top fit together with the blue boxes below? Should they match the 4 steps? Can we actually see quantification of the region in the figure?

Abbreviations have been fixed. The blue boxes didn't add anything so we removed them, and instead added an arrow indicating how a) relates to b), and also added another section to further clarify what StratoMod's predictions mean.

8. Figure 5 is missing – maybe a wrong label of Figure 6?

This has been fixed. We moved Fig 5 to the supplement and forgot to change Fig 6.

9. The resolution of the IGV screenshots should be improved.

We added bigger text to the screenshot in fig 3 to make this more readable.

10. The authors mention 22 features, but in Supp table 1 I only find 17? The categorical feature (seq platform) is not included. Regarding the other 4 – I don't know what they are.

All models (except the PCR-free/plus model) used 22 features. In Supp Table 1, only 15 rows apply to these models; the 2 rows that don't apply are for VCF_VAF and VCF_DP which are noted as only applying to the PCR-free/plus model. Of those 15 rows, 12 of them represent one

column in the dataset. Of the remainder, 2 rows are for homopolymer features (HOMOPOL_<base>_imperfect_frac and HOMOPOL_<base>_length) which were created for each base, and rather than write almost the same thing for each base we used the placeholder <base> to denote each of the 4 bases. Thus each of these rows actually represents 4 columns apiece. The final row in the remainder is for mappability (MAP_difficult_<X>.bp) for which the X stands for either 100 or 250bp, thus this row actually represents 2 features.

So in totality: $12 + 4 + 4 + 2 = 22$

Note that the PCR-free/plus model is described as using 24 features, which is the same as the above except VCF_VAF and VCF_DP are included.

“Seq platform” is denoted by VCF_input.

We made this more obvious by adding some clarifying phrases in both the methods (line 941 and 951) and supplemental table 1 (particularly the former since this is more likely to be read) and adding colors to Supp Table 1 to highlight the fact that the <base> and <X> are shorthand for several features.

11. A reference for BWA is missing. Do the authors use BWA mem? Same for vg (which is not spelled VG).

This was indeed BWA mem. We cited this as well as VG to clarify (line 385). As for the spelling of VG, its authors appear to use both (<https://github.com/vgteam/vg>).

12. As the authors use gnomAD 4.0, which was released in Nov 2023, they should not use a ClinVar release from Nov 2022.

As discussed above, while it is true that variants are constantly being added to the database, it isn't clear how simply adding more variants would change our conclusions. We don't think the mismatch in dates for gnomAD and ClinVar should cause any problems for the model, but we've now clarified in the discussion that StratoMod could be used to make predictions for new variants as they continue to be added to ClinVar, which may become more important once results from long reads are added to ClinVar (line 637).

13. When trying out StratoMod: The exemplary file is not called dynamic-testing.yml, but testing.yml. The authors should update this information in their git-repository. Unfortunately, I could not successfully execute the example, the error message was:

```
pip._vendor.urllib3.exceptions.ReadTimeoutError:  
HTTPSConnectionPool(host='files.pythonhosted.org', port=443): Read timed out. failed  
CondaEnvException: Pip failed
```

The typos have been fixed. Thank you for pointing these out.

We are not sure why you are getting this error. This appears to be a network issue with pip rather than a StratoMod issue. It may be possible that you are using conda/mamba versions which are vastly different from what we are using. We clarified which versions we used in the README.

Multiple people on our team (not directly involved in developing this project) have used this across different systems (Linux and Mac), and it also has worked for others who have contacted us about using StratoMod. If this persists, please open an issue on Github.

Dear reviewers,

Thank you for taking the time to review our manuscript entitled “StratoMod: Predicting sequencing and variant calling errors with interpretable machine learning”. Please see the following for our responses to your feedback.

Note that all line numbers refer to the main text. Our responses are in blue, occasionally referring to specific feedback in red.

Reviewer #2 (Remarks to the Author):

The authors have satisfactorily addressed my prior comments. I particularly appreciate the changes in phrasing that make it more clear what StratoMod is predicting. I have a few minor follow-up comments on the revised manuscript:

1. Figure 1c: I understand that one could predict the Jaccard Index, but I have hard time seeing any practical use of this and thus I suggest removing this scenario from the figure as it leads to confusion. In particular, to compute the jaccard index, one needs to predict at every position that is either in the benchmark or in the query - but if you know the positions in the benchmark and query you can directly compute the jaccard index without any prediction. I don't see a scenario in which you know that a position is in either the benchmark or query but do not know if it is in both.

I suggest removing this scenario from the figure as it leads to confusion

We actually use the Jaccard index in the "Model Validation" section, so we do not agree it should be removed from Figure 1c. However, this was likely not clear since we did not mention the phrase "Jaccard index" anywhere in this section. This has been fixed (line 457). We originally didn't use this term when describing the validation because it isn't a common term used by the genomics community nor is it commonly used in binary classification (at least in our experience).

We also added some introductory remarks to the top of the results explaining why we chose the Jaccard index (line 150); this rationale was already in the validation section so it is easier to compare to the other use cases throughout the text.

I don't see a scenario in which you know that a position is in either the benchmark or query but do not know if it is in both.

There are two ways in which StratoMod can be used: explainability and prediction (which may also include explainability). In the former case, one is using the model as means to understand how different genome contexts lead to a given outcome. This is analogous to linear regression, where one is most concerned about the slopes of the model. In this case, one already knows if a variant is in the benchmark or query due to both being present in the training set (to directly address this critique). Either in addition to or instead of this, one may be interested in predicting a variant that is outside the original training set. In this case, the prior probabilities we outlined in Supp Note 4 are important to understand, since they inform where this previously-unseen-by-the-model variant is likely to be seen in real life. We did not make this point before, thus added it starting at line 665.

The gnomAD comparison is an example of using StratoMod (albeit in a specialized way) to predict the Jaccard index of variants in gnomAD samples distinct from the data from the HG002 sample that was used for training. In gnomAD, the non-pass variants should be enriched for

either FP or FN errors. The former arise due to sequencing or mapping errors, and the latter arise from correct variants being called in regions that are difficult for short reads (segmental duplications, repeats, etc). By training StratoMod to predict both FN and FP errors with Illumina reads, we hypothesized that StratoMod's predictions should correlate with gnomAD, since the errors correspond to the same mechanisms (and we found evidence for this). While we did not do this here, StratoMod's explainability could also be utilized to understand which features lead to probability of overlap. Hopefully the role of the Jaccard index in this logic is more obvious now that we added more proper terminology.

2. Now that I better understand the section "Performance assessment and prediction of missing clinically-relevant variants", I believe there is still some confusing language in this section with regard to "precision" and "recall". In much of the rest of the manuscript, "precision" and "recall" refer to the variant calls of a particular *sequencing pipeline*. However, in this section, if I understand correctly, "precision" and "recall" refer to the predictions of *StratoMod* at ClinVar variant positions. Some clarification of this language would help the reader.

This is admittedly confusing since we have two levels of "precision and/or recall," one for StratoMod's output (which relates to variant benchmarking, which is described using precision or recall) and the other is for binary classification performance evaluation. We added extra clarification to this section (changes starting at line 253) and to the supplement (line 167 and many figure captions) to remedy this.

Reviewer #3 (Remarks to the Author):

Most of my comments were addressed. I just have one more remark: The authors commented on ClinVar containing only germline variants until 2024. This is not thoroughly correct. It is true that ClinVar recently introduced "an enhanced data model to better accept and support classifications of somatic variants". But: it contained somatic variants even before that. For example NRAS G12A (<https://www.ncbi.nlm.nih.gov/clinvar/variation/219097/>). It is in there since 2016 and a very common cancer variant. I have seen this variant many times in various data sets. It is a somatic variant, often present at low VAFs. According to a paper by Ritter et al. (Genome Medicine 2016), ClinVar contained - at that time - around 2% somatic variants. Maybe the authors could comment on this? As it affects their approach, especially the use of DeepVariant and the focus on not-WES and not-targeted seq data.

There are two components to this.

First, this is not a concern regarding our ClinVar/gnomAD comparison we used for model validation. In this case, we intersected (likely) pathogenic variants with gnomAD variants (from genomic data only), and only considered those that were in both. Since gnomAD is a database of germline variants, this implies the only variants we considered from ClinVar were also germline.

Second, it is true that we are likely including a few somatic variants in our Hifi vs Illumina analysis (Figure 3). However, this is likely not a concern given the low percentage. In Figure 3a the two platforms differ by over an order of magnitude, and removing 2% of these will likely change the outcome little. In Figure 3b, almost all the data points (including the assumed somatic ones) follow the same trend (Hifi is more confident than Illumina) so removing these would not change our conclusion. In Figure 3b, the case we are showing is likely a germline variant due to the allele fraction in the Hifi track (50%). Furthermore, the goal of using StratoMod to predict ClinVar false negatives was to showcase how it can be used to make predictions about a representative (albeit not perfect) callset which is important to a wider audience. In reality, anyone using StratoMod would likely have their own variants they care about measuring, and in that case they would likely know if they are somatic or not.

We addressed both these concerns in the discussion (starting in line 657).